# EditScore: Unlocking Online RL for Image Editing via High-Fidelity Reward Modeling

**Xin Luo**[1,3][*], **Jiahao Wang**[2,3][*], **Chenyuan Wu**[1,3][*], **Shitao Xiao**[3], **Xiyan Jiang**[3,4],
**Defu Lian**[1], **Jiajun Zhang**[2], **Dong Liu**[1], **Zheng Liu**[3,5][†]

[1] University of Science and Technology of China
[2] Institute of Automation, Chinese Academy of Sciences
[3] Beijing Academy of Artificial Intelligence, [4] Zhejiang University
[5] Hong Kong Polytechnic University

 GitHub: https://github.com/VectorSpaceLab/EditScore

## ABSTRACT

Instruction-guided image editing has achieved remarkable progress, yet current models still face challenges with complex instructions and often require multiple samples to produce a desired result. Reinforcement Learning (RL) offers a promising solution, but its adoption in image editing has been severely hindered by the lack of a high-fidelity, efficient reward signal. In this work, we present a comprehensive methodology to overcome this barrier, centered on the development of a state-of-the-art, specialized reward model. We first introduce **EditReward-Bench**, a comprehensive benchmark to systematically evaluate reward models on editing quality. Building on this benchmark, we develop **EditScore**, a series of reward models (7B–72B) for evaluating the quality of instruction-guided image editing. Through meticulous data curation and filtering, EditScore effectively matches the performance of learning proprietary VLMs. Furthermore, coupled with an effective self-ensemble strategy tailored for the generative nature of EditScore, our largest variant even surpasses GPT-5 in the benchmark. We then demonstrate that a high-fidelity reward model is the key to unlocking online RL for image editing. Our experiments show that, while even the largest open-source VLMs fail to provide an effective learning signal, EditScore enables efficient and robust policy optimization. Applying our framework to a strong base model, OmniGen2, results in a final model that shows a substantial and consistent performance uplift. Overall, this work provides the first systematic path from benchmarking to reward modeling to RL training in image editing, showing that a high-fidelity, domain-specialized reward model is the key to unlocking the full potential of RL in this domain. Our code, models, data and benchmark will be released publicly.

## 1 Introduction

Recently, Reinforcement Learning (Li, 2017) has demonstrated immense power in advancing large language models and has also shown remarkable success in the text-to-image (T2I) domain. Works like FlowGRPO (Liu et al., 2025a) and DanceGRPO (Xue et al., 2025) have leveraged RL on flow-matching models to significantly enhance T2I generation across multiple dimensions. Despite these successes, the application of RL to the image editing domain remains largely underexplored. We posit that RL holds significant, untapped potential for image editing. By framing editing as a dynamic trial-and-error process, a policy model can discover novel and more effective editing strategies beyond what is captured in static datasets. Through a well-designed reward mechanism, the model can be progressively steered to better align with user intent and achieve a deeper, more robust editing capability across diverse scenarios.

---

[*]Equal Contribution.
[†]Corresponding Author.

Despite its theoretical promise, applying online RL to high-resolution image editing remains a formidable and largely unsolved challenge (Wei et al., 2025; Wu et al., 2025a; Ahmadi et al., 2025). The primary obstacle lies in the absence of a suitable reward function: a reliable, efficient, and scalable oracle that can accurately score the quality of an edit given an instruction. A natural consideration would be to employ state-of-the-art, general-purpose Visual Language Models (VLMs) like GPT-5 (OpenAI, 2025) or other proprietary APIs as reward providers. However, this approach is impractical for online RL, with these large VLMs are expensive to query at scale. The second alternative, using powerful open-source VLMs, resolves cost issues. However, our investigations reveal a critical performance gap. As we will demonstrate, even Qwen2.5-VL-72B (Bai et al., 2025b) fails to provide a sufficiently accurate and consistent reward signal to effectively guide the policy. Directly using such models as reward functions often causes unstable training or policy collapse, showing that scale alone cannot replace specialized accuracy. Progress thus hinges on an accurate reward model tailored to image editing, yet the community still lacks a powerful open-source option—posing a major barrier to advancing RL-based editing.

In this work, we argue that the key to unlocking online RL for image editing is the development of a specialized, high-fidelity, and efficient reward model. We present a systematic approach to build, validate, and deploy such a model. Our contributions are organized as follows:

First, to enable rigorous and reproducible research, we introduce **EditReward-Bench**, a comprehensive benchmark for evaluating reward models in image editing. It is organized into four main categories, comprising 13 diverse subtasks ranging from simple attribute edits to complex compositional changes. Each entry is evaluated by expert human raters across three key dimensions: prompt following, consistency, and overall quality. All annotations have passed rigorous inter-annotator agreement checks, establishing a reliable standard for measuring reward model quality.

Second, guided by our benchmark, we develop **EditScore**, a series of powerful reward models (7B–72B) for evaluating the quality of instruction-guided editing. Through careful data curation and filtering and inference-time scaling, EditScore sets a new state of the art for open-source reward models in this domain, even surpassing the accuracy of leading proprietary VLMs.

To validate its practical utility, we first apply EditScore in a Best-of-$N$ selection experiment across several state-of-the-art editing models, which confirms that our reward model can effectively improve the performance of diverse editors. We further used EditScore for RL training, resulting in a model that shows a substantial and consistent performance uplift over its base model.

In summary, our key contributions are as follows.

- We propose EditReward-Bench, a new public benchmark for the direct and reliable evaluation of reward models for instruction-guided image editing.
- We develop and release EditScore, a series of state-of-the-art open-source reward models for instruction-based image editing that, through our self-ensembling strategy, surpasses the accuracy of leading proprietary VLMs on this task.
- We demonstrate the broad, practical utility of EditScore through a Best-of-$N$ selection experiment, successfully enhancing the performance of multiple diverse base models.
- We provide a comprehensive analysis of an end-to-end online RL application, identifying key factors for success, including the accuracy and variance of the reward signal.

## 2 RELATED WORK

**Reward Models for Image Generation.** Reward models provide signals based on specific preferences and can be applied in various scenarios such as automated evaluation and reinforcement learning. Most research has focused on text-to-image tasks, with works like Wu et al. (2023b); Kirstain et al. (2023); Zhang et al. (2024b) fine-tune CLIP-based model with human preference data and Ma et al. (2025); Wang et al. (2025b); Gao et al. (2025) employ VLMs as a core component for reward generation. Research on reward models for image editing is relatively underexplored: some works focused on controlled edits based on predefined masks (Ren et al., 2024; Gong et al., 2025). For instruction-guided editing, Zhang et al. (2024a) fine-tunes the BLIP model (Li et al., 2022) with human annotated reward signals for the image generated only by P2P Hertz et al. (2022) and IP2P Brooks et al. (2023), while Chen et al. (2025) leverages CLIP scores to automatically construct

training data for training LLaVA-Next-8B Liu et al. (2024) as reward model. However, they used outdated editing models to generate images that needed to be evaluated, or simply used CLIP to annotate and score the images. Meanwhile, they use limited task categories as the prompt pool. The limitations in the narrow scope of task and generation model coverage make them struggle to support online RL for current models. Furthermore, they are not open-source and cannot be used and evaluated by the public. In contrast, our model enables broader evaluation of editing tasks (13 tasks) while supporting SOTA editing models (e.g., Gemini-2.5-image-preview). Moreover, its generative nature enables it to perform inference-time scaling to enhance scoring accuracy (Liu et al., 2025d).

**Image Reward Model Evaluation.** Establishing a reliable benchmark is critical as it ensures the objective measurement of reward modeling performance and offers clear guidance for optimization. In previous work, image reward models are evaluated on their accuracy in predicting human preferences for key generative tasks, such as text-to-image generation (Xu et al., 2023; Kirstain et al., 2023; Wu et al., 2023b;a; Ma et al., 2025) and image editing (Ku et al., 2023b; Jiang et al., 2024). Over the past year, generative models such as GPT-Image-1 (Hurst et al., 2024), FLUX-Kontext (Batifol et al., 2025), Qwen-Image-Edit (Wu et al., 2025a) and Nano Banana (Google, 2025) have achieved significant breakthroughs in image editing, particularly in stylization, hybrid edit and text modification. With the growing power of generative models and their expanding editing capabilities, there is an urgent need for reward models that can accurately assess them. This, in turn, necessitates the development of a novel and more comprehensive benchmark for evaluating edit rewards.

# 3 EDITREWARD-BENCH

## 3.1 OVERVIEW

EditReward-Bench is a benchmark specifically designed for systematic evaluation of reward modeling for image editing. It covers 13 diverse editing tasks and spans 11 heterogeneous editing models for data construction, ranging from open-source baselines to state-of-the-art proprietary models (see Table 1 for comparison). The benchmark is distinguished by the following features.

**Multi-dimensional Image Evaluation Framework.** EditReward-Bench offers three key dimensions for evaluating editing outcomes. These includes Prompt Following (adherence to prompts), consistency (preservation of key visual elements) and overall quality (comparison across all aspects).

**Comprehensive Model Performance Spectrum.** EditReward-Bench incorporates both state-of-the-art editing models and lower-peforming baselines. It effectively challenges the reward model's distinguish ability and validates the reward model's scoring performance on current SOTA models.

**Extensive Task Coverage with Real-world Applicability.** EditReward-Bench also covers a diverse set of editing tasks that closely align with real-world application scenarios, ensuring authentic and comprehensive evaluation of reward models for image editing.

Table 1: Comparison of **EditReward-Bench** against existing benchmarks, highlighting its superior scale, data sources, and comprehensive task coverage. The order of subtasks within each tuple under "Task Coverage" corresponds to the order of subtasks listed for each category in Section 3.2.

| Feature | ImagenHub (Ku et al., 2023b) | GenAI-Bench (Jiang et al., 2024) | **EditReward-Bench (ours)** |
|---|---|---|---|
| *General Properties* | | | |
| Size | 2,864 | 919 | **3,072** |
| Multi-Dimensional Eval. | ✓ | ✗ | ✓ |
| Proprietary Model Data | ✗ | ✗ | ✓ |
| *Task Coverage (Conceptual Groups)* | | | |
| Subject | (✓,✓,✓) | (✓,✓,✓) | (✓,✓,✓) |
| Appearance | (✓,✓,✗,✗) | (✓,✓,✗,✗) | (✓,✓,✓,✓) |
| Scene | (✓,✗) | (✓,✗) | (✓,✓) |
| Advanced | (✗,✓,✓,✗) | (✗,✓,✓,✗) | (✓,✓,✓,✓) |

## 3.2 CONSTRUCTION PROTOCOL

To create a robust and comprehensive benchmark, we focused on three key pillars: the diversity of editing tasks, the variety of editing models, and the granularity of our evaluation criteria.

First, to ensure comprehensive task coverage, we structure our benchmark into four main categories, comprising 13 distinct subtasks curated from established datasets like GEdit-Bench-EN (Liu et al., 2025b) and ImgEdit-Bench (Ye et al., 2025). These categories are designed to span a wide spectrum of complexity. They are: 1) **Subject**, which includes fundamental tasks like `subject addition`, `subject removal` and `subject replace`; 2) **Appearance**, covering shape-preserving edits such as `color alteration`, `material modification`, `style transfer` and `tone transformation`; 3) **Scene**, which tests understanding of image layout through tasks like `background change` and `extract`; and 4) **Advanced**, the most challenging category, which requires advanced reasoning for tasks like `portrait beautification`, `text modification`, `motion change` and `hybrid edit`.

Second, to populate our benchmark with a diverse distribution of edited images, the candidate pool of outputs were generated by a diverse array of 11 generative models for data construction: Step1X-Edit (Liu et al., 2025b), Step1X-Edit v1.1 (Liu et al., 2025b), Qwen-Image-Edit (Wu et al., 2025a), OmniGen2 (Wu et al., 2025b), FLUX-Kontext-dev (Batifol et al., 2025), FLUX-Kontext-pro (Batifol et al., 2025), Bagel (Deng et al., 2025), MagicBrush (Zhang et al., 2023), Omnigen (Xiao et al., 2025), gpt-image-1 (Hurst et al., 2024), and Gemini-2.5-image-preview (Google, 2025), including both open-source and state-of-the-art proprietary editors.

Finally, recognizing that the quality of editing results is not monolithic, we designed a multi-dimensional evaluation scheme for better interpretability as inspired by (Ku et al., 2023a; Liu et al., 2025b). Each editing result is assessed along three distinct axes: **Prompt Following** (PF): measuring how faithfully the edit executes the given instruction. **Consistency** (C): assessing the preservation of unedited image regions. **Overall Quality** (O): providing a holistic comparison of edits by accounting for all relevant aspects.

## 3.3 ANNOTATION PIPELINE

To ensure the reliability of EditReward-Bench, we designed a rigorous human annotation pipeline conducted exclusively by experts in generative AI, as illustrated in Figure 1.

To ensure high-quality ground truth, we implemented a **Two-Annotator Discussion Protocol**. Unlike standard crowdsourcing where raters work in isolation, our protocol assigns two experts to each sample. They engage in real-time discussion to analyze visual artifacts (e.g., attribute drift) and resolve discrepancies, finalizing the ranking only upon reaching a joint consensus. We validated this protocol through a controlled study comparing it against independent labeling. The results demonstrate that the discussion-based approach significantly reduces annotation noise: for instance, in the Consistency dimension, the inter-annotator convergence rate (at a 100% agreement threshold) improved by **12.12%**. Furthermore, the rankings determined by a single annotator pair achieved over **97%** consistency with the majority vote derived from the full expert pool, confirming that our efficient two-annotator setup yields results highly aligned with collective expert judgment. A detailed disagreement analysis is provided in Appendix B.

Following this protocol, for each input with five output images randomly sampled from the candidate pool, the annotators were asked to rank them. Our system uniquely allowed them to group outputs of similar quality into the same tiers; for instance, a ranking of 3|12|45 places output 3 in the top tier, outputs 1 and 2 tied in the second, and 4 and 5 tied in the third.

These tiered rankings were then decomposed into pairwise preference tuples. Following the 3|12|45 example, this conversion yields preference pairs where a higher-tiered item is preferred over a lower-tiered one, such as $(3, 1)$, $(3, 2)$, $(1, 4)$ and $(2, 5)$. Comparisons between items within the same tier (e.g., $(1, 2)$) are excluded as they represent equal quality. This process resulted in a large-scale, high-fidelity dataset of **3,072** preference pairs, comprising 944 for prompt following, 890 for consistency, and 1,238 for overall quality. For the evaluation of reward models, we utilize the preference prediction accuracy metric, calculated by the proportion of pairs $(A, B)$ where the model scores the human-preferred output higher, i.e., $S(A) > S(B)$.

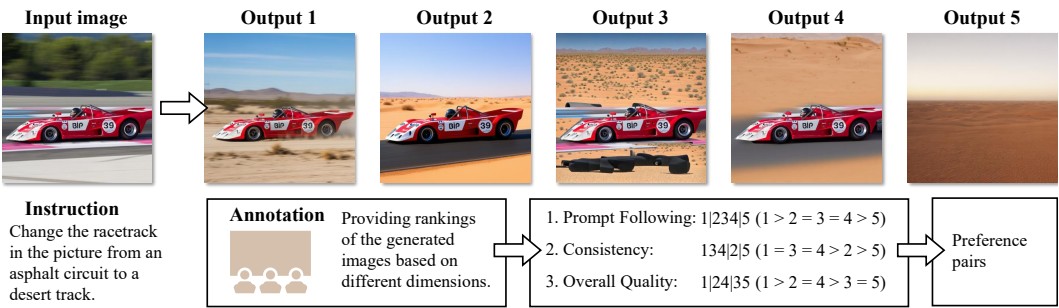

Figure 1: Illustration of the annotation process. Annotators are presented with five candidate output images and are asked to rank them according to three evaluation dimensions. The final ranking is determined through consensus among multiple annotators. For example, 1|234|5 indicates that the first image is preferred over images 2, 3 and 4, which are in turn preferred over image 5.

## 4 METHOD

### 4.1 EDIT REWARD

#### 4.1.1 THE APPROACH TO REWARD MODELING

Inspired by (Wang et al., 2025b;a; Wu et al., 2025c), we formulate reward modeling as a conditional textual generation task. We fine-tune models from the powerful Qwen2.5-VL series (Bai et al., 2025b) on a standard autoregressive objective to act as specialized evaluators. Our model **EditScore** takes (Instruction, Input Image, Output Image) as input, (Reasoning, Scalar Score) as output. The chain-of-thought (Wei et al., 2022) process of output not only enhances the model's interpretability but also demonstrably improves the accuracy of the final scores.

To ensure EditScore can perform a comprehensive evaluation of editing tasks, we adopt the VI-EScore framework (Ku et al., 2023a), prompting the model in parallel to assess two orthogonal aspects. The first is **Semantic Consistency (SC)**, which evaluates the degree of instruction following and consistency, ensuring specified objects are correctly modified while unmentioned regions remain preserved. The second is **Perceptual Quality (PQ)**, which assesses overall image quality, focusing on photorealism and the absence of artifacts. The final score is the geometric mean of the SC and PQ scores ($S_{final} = \sqrt{S_{SC} \cdot S_{PQ}}$), yielding a balanced and granular reward signal.

#### 4.1.2 INFERENCE-TIME ENSEMBLING STRATEGY

Recent work (Snell et al., 2024; Liu et al., 2025c) has shown that the performance of LLMs can be substantially improved by increasing the computational budget at test time. Building on this insight, we introduce an inference-time ensembling strategy for our generative reward model to systematically boost its evaluation accuracy and enhance reward stability. Our approach is straightforward: for a given input triplet $\mathbf{z} = $ (Instruction, Input Image, Output Image), we perform $K$ independent, stochastic forward passes through EditScore, and each pass $i$ generates a pair (reasoning$_i$, $s_i$). Then we aggregate only the scalar scores $\{s_1, s_2, \ldots, s_K\}$ to compute the final ensembled score, $S_{final}$, using the arithmetic mean:

$$S_{final}(\mathbf{z}) = \frac{1}{K} \sum_{i=1}^{K} s_i, \quad \text{where } (\text{reasoning}_i, s_i) = \text{EditScore}(\mathbf{z}). \tag{1}$$

An intuitive explanation (Liu et al., 2025c) for this effectiveness is that each of the $K$ generated reasonings can be viewed as distinct judgment perspectives. Aggregating the scores derived from these diverse perspectives allows the final reward to more accurately reflect the true quality of an edit, leading to significant scaling effectiveness.

## 4.2 Data Construction Pipeline

The training data for reward model and subsequent reinforcement learning are constructed following the similar procedure. For reinforcement learning, the training data only requires the input images and instructions, whereas training the reward model additionally requires the generated outputs paired with their corresponding rewards. The construction process involves three steps as follows.

**Step I: Images selection and instructions creation.** We start by selecting and filtering a diverse set of high-quality images as editing inputs. Next, we construct a series of reference instructions for various editing tasks. Using the input images and random selected reference instructions as guidance, we prompt Qwen-2.5-VL-72B to generate editing instructions that are task-consistent with the references. After generating a large set of image-instruction pairs, we applying a K-center greedy algorithm (Sener & Savarese, 2018) to select 1000 semantically diverse samples per task for reward generation.

Finally, we constructed 70,000 data samples for training the reward model and 60,000 samples for reinforcement learning training. The data for the reward model requires further annotation as follows.

**Step II: Candidates output generation.** We generate candidates output using 5 distinct editing models random selected from our model pool.

**Step III: Annotate and filtering.** For each candidate output, we use GPT-4.1 to annotate scores. Following VIEScore (Ku et al., 2023a), we score each output on SC and PQ and provide reasons for the assigned scores. Next, we apply filtering to the reward samples. It is conducted from two perspectives: (i) filtering based on group-wise maximum scores to remove unachievable editing tasks, and (ii) filtering by group standard deviation to remove cases with low discriminability.

## 5 Reward Model Performance on EditReward-Bench

### 5.1 Experimental Setup

Our final EditScore model is obtained by fine-tuning Qwen2.5-VL with LoRA (Hu et al., 2022) on our curated dataset (see Section 4.2). For evaluation, we adopt the VIEScore (Ku et al., 2024) prompt template with two key modifications: (i) enforcing a reasoning-before-scoring format, (ii) expanding the score range from $[0, 10]$ to $[0, 25]$. Following the formulation of VIEScore, we derive Prompt Following ($S_{PF}$) and Consistency ($S_C$) from its Semantic Consistency metric $S_{SC}$, while Overall Quality ($O$) is directly taken from the final score $S_{final}$. The detailed prompt templates are shown in Appendix M.

### 5.2 Main Results

Results summarized in Table 2 reveal a significant performance gap between proprietary models and open-source counterparts. Leading proprietary models such as GPT-4.1, GPT-5 and Gemini-2.5-Pro form a distinct upper tier, achieving pairwise accuracies in the 0.7-0.75 range across all dimensions. This confirms their strong, albeit imperfect, zero-shot capabilities for this nuanced task. In particular, we find that VLMs are generally stronger at assessing Prompt Following (PF) than Consistency (C), as the latter requires fine-grained comparisons between the input and output images.

In stark contrast, even the largest and most capable open-source models exhibit notable limitations. The Qwen2.5-VL series shows a clear scaling trend, yet the 72B-parameter variant still falls short of 0.612 overall accuracy and performs worse than random chance in Consistency judgment. The smaller 7B and 32B models fare even worse, underscoring the inadequacy of off-the-shelf open-source VLMs as reliable reward signals for fine-grained editing tasks. By contrast, our EditScore achieves substantial improvements: the 7B variant surpasses the 10x larger Qwen2.5-VL-72B, while the 72B variant matching the score of GPT-4.1. Moreover, scaling inference-time compute with self-ensemble (Avg@4) further boosts performance across all model sizes, with EditScore-72B establishing the state of the art on EditReward-Bench.

Table 2: Benchmark results on **EditReward-Bench**, reporting both overall pairwise accuracy and a fine-grained breakdown across four categories of edit capabilities. Pairwise accuracy measures the proportion of pairs where the model correctly assigns a higher reward score to the human-preferred output. Notably, **EditScore** achieves superior performance even with its compact 7B size. Avg@4 denotes the average score over 4 forward passes.

| | Model Metric | GPT-4.1 | GPT-5 | Gemini-2.5 Pro | Qwen2.5-VL 7B | Qwen2.5-VL 32B | Qwen2.5-VL 72B | EditScore-7B Base | EditScore-7B Avg@4 | EditScore-32B Base | EditScore-32B Avg@4 | EditScore-72B Base | EditScore-72B Avg@4 |
|---|---|---|---|---|---|---|---|---|---|---|---|---|---|
| **Overall** | PF | 0.673 | **0.777** | 0.703 | 0.458 | 0.498 | 0.540 | 0.592 | 0.722 | 0.638 | 0.736 | 0.635 | 0.755 |
| | C | 0.602 | 0.669 | 0.560 | 0.325 | 0.376 | 0.435 | 0.591 | 0.720 | 0.556 | 0.704 | 0.586 | **0.735** |
| | O | 0.705 | 0.755 | 0.722 | 0.432 | 0.563 | 0.621 | 0.659 | 0.727 | 0.680 | 0.733 | 0.703 | **0.763** |
| **Subject** | PF | 0.615 | 0.707 | **0.712** | 0.414 | 0.527 | 0.523 | 0.590 | 0.691 | 0.625 | 0.703 | 0.612 | 0.708 |
| | C | 0.520 | 0.538 | 0.465 | 0.317 | 0.414 | 0.394 | 0.585 | **0.666** | 0.473 | 0.627 | 0.524 | 0.639 |
| | O | 0.679 | 0.708 | 0.765 | 0.460 | 0.570 | 0.594 | 0.740 | 0.771 | 0.703 | 0.754 | 0.721 | **0.807** |
| **Appear.** | PF | 0.673 | **0.762** | 0.631 | 0.422 | 0.393 | 0.390 | 0.573 | 0.682 | 0.587 | 0.714 | 0.584 | 0.733 |
| | C | 0.668 | 0.714 | 0.577 | 0.335 | 0.320 | 0.416 | 0.612 | 0.730 | 0.591 | 0.764 | 0.623 | **0.778** |
| | O | 0.709 | **0.756** | 0.700 | 0.470 | 0.514 | 0.559 | 0.663 | 0.714 | 0.669 | 0.710 | 0.697 | 0.736 |
| **Scene** | PF | 0.763 | 0.852 | 0.766 | 0.433 | 0.611 | 0.690 | 0.744 | 0.821 | 0.789 | 0.870 | 0.788 | **0.908** |
| | C | 0.682 | 0.741 | 0.675 | 0.236 | 0.482 | 0.429 | 0.735 | **0.835** | 0.627 | 0.787 | 0.695 | 0.797 |
| | O | 0.773 | **0.841** | 0.693 | 0.357 | 0.673 | 0.713 | 0.789 | 0.774 | 0.764 | 0.816 | 0.794 | 0.837 |
| **Advanced** | PF | 0.673 | **0.806** | 0.736 | 0.541 | 0.524 | 0.627 | 0.536 | 0.736 | 0.625 | 0.717 | 0.625 | 0.734 |
| | C | 0.556 | 0.687 | 0.557 | 0.367 | 0.351 | 0.488 | 0.503 | 0.693 | 0.548 | 0.658 | 0.541 | **0.733** |
| | O | 0.686 | **0.746** | 0.724 | 0.410 | 0.553 | 0.657 | 0.529 | 0.683 | 0.631 | 0.699 | 0.650 | 0.721 |

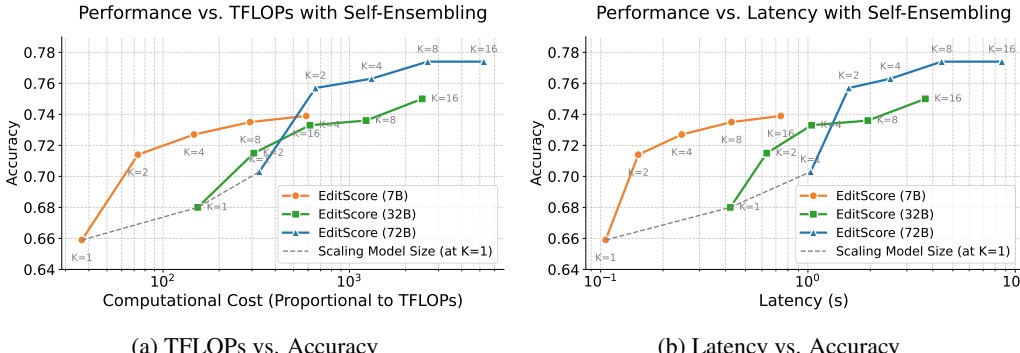

(a) TFLOPs vs. Accuracy  (b) Latency vs. Accuracy

Figure 2: **Self-ensembling offers a superior efficiency-performance trade-off compared to simply scaling model parameters. (a) TFLOPs vs. Accuracy:** The colored solid lines (increasing $K$) show a steeper performance trajectory than the gray dashed line (scaling model size), indicating a higher return on computational investment. **(b) Latency vs. Accuracy:** Real-world measurements confirm that the latency cost of self-ensembling is sublinear due to shared KV-cache prefilling. Notably, **EditScore-7B** ($K = 4$) achieves higher accuracy than the single-pass **EditScore-32B** ($K = 1$) while requiring significantly less latency and hardware resources.

## 5.3 EFFECTIVE INFERENCE-TIME SCALING OF EDITSCORE

As shown in Table 2, both scaling model size and inference-time compute improve the performance of EditScore. To further investigate which dimension contributes more effectively, we normalize the comparison by using FLOPs as a proxy for inference cost. Figure 2 analyzes the scaling properties of EditScore from two perspectives: theoretical computational cost (TFLOPs, Left) and real-world inference latency (Right). In both plots, colored solid lines denote inference-time compute scaling (increasing ensemble size $K$), while the gray dashed lines represent parameter scaling (increasing model size from 7B to 72B). We observe that **scaling inference-time compute yields significantly greater marginal gains** than scaling model parameters. Moreover, as shown in Figure 2(b), the wall-clock latency grows sublinearly with $K$. This efficiency stems from the shared **prefill stage** across ensemble members within the attention mechanism, minimizing redundant computation. For a comprehensive breakdown of memory consumption, system throughput, and normalized hardware budgets, please refer to Appendix C.

## 5.4 Ablation Study

We study two key factors in EditScore design: score range granularity and output format (Table 3). Increasing the target score range generally improves pairwise accuracy for both GPT-4.1 and GPT-5, peaking around $[0, 25]$, while overly large ranges hurt performance due to regression difficulty. In parallel, requiring the model to generate a rationale before the numeric score ("reasoning + score") consistently outperforms direct scoring, yielding a +0.038 accuracy gain for EditScore-7B ($0.621 \rightarrow 0.659$). These findings highlight that both an appropriately chosen score range and reasoning-first output are crucial for maximizing accuracy.

Table 3: The effect of score range granularity and reason first on the pairwise accuracy of VLMs on EditReward-Bench. The ranges ($[0, x]$) indicate the target score space for model outputs. The lightly shaded column highlights our chosen configuration for optimal performance.

| Method | Score Range | | | | Output Format | |
|---|---|---|---|---|---|---|
| | $[0, 10]$ | $[0, 20]$ | $[0, 25]$ | $[0, 30]$ | reasoning + score | score |
| GPT-4.1 | 0.691 | 0.701 | 0.705 | 0.689 | 0.705 | 0.695 |
| GPT-5 | 0.730 | **0.760** | 0.755 | — | 0.755 | 0.741 |
| EditScore-7B (GPT-4.1) | 0.605 | 0.657 | 0.659 | 0.629 | 0.659 | 0.621 |

## 6 Application of EditScore in Image Editing

### 6.1 Experimental Setup

**Evaluation Method**. To evaluate the effectiveness of EditScore in improving image editing models, we design two experiments. (1) best-of-$N$ selection: EditScore is used as a selector, where the editing model generates multiple candidate outputs per input and EditScore chooses the best one. We evaluate the gain on three popular models: OmniGen2 (Wu et al., 2025b), Flux.1-Kontext-dev Batifol et al. (2025) and Qwen-Image-Edit Wu et al. (2025a). (2) Reinforcement learning: EditScore is employed directly as a reward model to fine-tune OmniGen2 via an additional RL stage, demonstrating its utility as a training signal. We adopt two widely used image editing benchmarks—GEdit-Bench (Liu et al., 2025b) and ImgEdit-Bench (Ye et al., 2025)—which cover a diverse range of practical editing tasks, to assess the improvements brought by EditScore. For efficiency, we use **7B variant** of EditScore for experiments with optional self-ensemble.

**RL Training.** Our main experiments are conducted using OmniGen2 Wu et al. (2025b), while we also validate the generalization of our method on Flux-Kontext-dev Batifol et al. (2025). Flow-GRPO Liu et al. (2025a) is employed with hyperparameters: sampling steps $T = 20$, group size $G = 12$, number of unique prompts $= 24$, noise level $\sigma = 0.9$, and KL weight $\beta = 0.04$. Ablation studies on hyperparameters are provided in Appendix E. We also reformulate the standard Flow-GRPO equation to align with OmniGen2's notation; detailed derivations are given in Appendices F and G.

### 6.2 Validating EditScore Utility via best-of-$N$ Section

Before implementing full RL training, we first validate the utility of EditScore in a controlled setting. In this experiment, the editing model generates $N$ candidate outputs per input of GEdit-Bench, and EditScore selects the best one. This setup directly measures the reward model's ability to identify high-quality edits. Figure 3a reports benchmark performance as a function of $N$. We evaluate three base models using both single-pass EditScore (solid lines) and the stronger Avg@4 version (dashed lines). The results highlight three key findings: (1) stronger reward models consistently yield better selections, (2) performance gains vary by base model, with Qwen-Image-Edit showing the least improvement due to prior post-training stage (Wu et al., 2025a), and (3) OmniGen2 exhibits the largest absolute potential for improvement, motivating its choice as the base model for subsequent RL experiments.

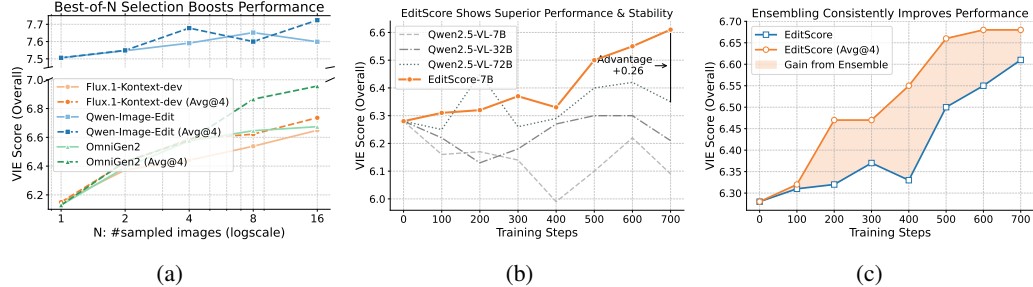

Figure 3: **EditScore as a superior reward signal for image editing.** (a) Using EditScore to select the best sample among multiple outputs effectively improves VIEScore, with OmniGen2 showing the largest gain. (b) Incorporating EditScore into RL training yields stable and significant performance improvements, even surpassing the much larger Qwen2.5-VL-72B. (c) RL training benefits from self-ensembling, which enhances the evaluation accuracy of EditScore across diverse settings.

Table 4: **Controlled Evaluation of Reward Models.** Using OmniGen2 as the fixed base, we compare policies trained with different reward signals. **EditScore** variants consistently outperform both raw VLMs and GPT-4.1, even on the out-of-distribution (OOD) EmuEdit benchmark. Note that raw Qwen2.5-VL baselines fail to provide effective guidance.

| Reward Signal | GEdit-Bench | | | ImgEdit | EmuEdit (OOD) | | |
|---|---|---|---|---|---|---|---|
| | SC | PQ | Overall | Overall | CLIP-I | CLIP-O | DINO |
| *Baseline (No RL)* | 6.72 | 7.20 | 6.28 | 3.40 | 0.876 | 0.309 | 0.822 |
| RL w/ Qwen2.5-VL-7B | 6.87 | 6.86 | 6.22 | - | - | - | - |
| RL w/ Qwen2.5-VL-72B | 6.89 | 7.21 | 6.42 | 3.60 | 0.871 | 0.311 | 0.814 |
| RL w/ GPT-4.1 | 7.24 | 7.41 | 6.73 | **3.66** | 0.884 | **0.312** | 0.843 |
| **RL w/ EditScore-7B (Avg@4)** | 7.20 | 7.46 | 6.68 | 3.63 | **0.900** | 0.310 | **0.870** |
| **RL w/ EditScore-Qwen3-VL-8B (Avg@4)** | **7.27** | **7.77** | **6.89** | 3.62 | 0.891 | 0.310 | 0.861 |

## 6.3 ONLINE REINFORCEMENT LEARNING WITH EDITSCORE

We now turn to the core application of our framework: using EditScore as a direct reward signal for online RL fine-tuning. This section systematically isolates the impact of reward quality, examines training stability, and evaluates generality across models and algorithms. We additionally provide qualitative visual comparisons in Appendix I.

### 6.3.1 UNLOCKING STABLE RL WITH A HIGH-FIDELITY REWARD SIGNAL

To isolate the contribution of the reward signal, we conducted a controlled study using **Omni-Gen2** as the policy model while varying the source of supervision. As summarized in Table 4, raw instruction-tuned VLMs prove ineffective as verifiers; even the large-scale **Qwen2.5-VL-72B** fails to provide meaningful guidance, resulting in unstable optimization trajectories (see Figure 3b). This underscores that parameter scale alone cannot substitute for task-specific alignment. In contrast, our specialized **EditScore-7B** not only stabilizes training but also demonstrates competitiveness with the proprietary **GPT-4.1**, surpassing it on the out-of-distribution EmuEdit benchmark (e.g., 0.900 vs. 0.884 CLIP-I). Furthermore, our approach benefits directly from stronger backbones: **EditScore-8B** (adapted from Qwen3-VL-8B (Bai et al., 2025a)) achieves the highest overall performance across metrics, confirming the method's scalability with advancing VLMs.

Beyond the reward model, we further investigate the impact of test-time compute. Figure 3c compares policy learning curves guided by a single-pass EditScore versus a 4-pass self-ensemble strategy (Avg@4). The ensembled signal significantly improves evaluation accuracy, leading to faster convergence and superior final performance. As we discussed in Appendix L, single pass evaluation occasionally produces unstable reward signal, while self-ensemble alleviate it, thus providing a consistent gradient for the policy to traverse.

Table 5: **Generality Analysis.** *Left:* Consistent improvements on a different backbone (Flux-Kontext-dev). *Right:* Compatibility with advanced RL algorithms (TempFlow-GRPO (He et al., 2025)).

<table>
<tr><td colspan="4" align="center">(a) Flux-Kontext-dev Backbone</td><td colspan="4" align="center">(b) Algorithm Comparison (OmniGen2 base)</td></tr>
<tr><td>**Method**</td><td>**SC**</td><td>**PQ**</td><td>**O**</td><td>**Algorithm**</td><td>**SC**</td><td>**PQ**</td><td>**O**</td></tr>
<tr><td>Baseline</td><td>6.59</td><td>7.61</td><td>6.15</td><td>Flow-GRPO</td><td>7.20</td><td>7.46</td><td>6.68</td></tr>
<tr><td>**RL w/ Ours**</td><td>**7.16**</td><td>**7.95**</td><td>**6.87**</td><td>**TempFlow-GRPO**</td><td>**7.53**</td><td>**7.99**</td><td>**7.21**</td></tr>
</table>

## 6.4 GENERALITY ACROSS EDITORS AND ALGORITHMS

We further assess the robustness of **EditScore-7B (Avg@4)** across different generative backbones and RL algorithms (Table 5). To verify model-agnostic generality, we apply it to a substantially different model, Flux-Kontext-dev (Batifol et al., 2025), observing strong improvements across all metrics (+0.57 SC, +0.72 Overall). Furthermore, substituting standard Flow-GRPO with TempFlow-GRPO (He et al., 2025)—which uses temporally aware loss reweighting—delivers further gains (Overall score of 7.21). This confirms that our reward signal is universally applicable and scales robustly with advanced RL algorithms without causing reward hacking.

## 6.5 ANALYSIS OF ANNOTATION SOURCES AND REWARD VARIANCE

Table 6: **Comparison of reward models based on annotator source.** We analyze reward models (RMs) trained on data from two different annotators: GPT-4.1 and GPT-5. The resulting policy's performance is evaluated by both annotators on GEdit-Bench (Liu et al., 2025b).

| Annotator | Component | Reward Model Metrics | | Policy Performance ↑ | |
|---|---|---|---|---|---|
| | | Acc. (O) ↑ | Score Std. | GPT-4.1 Score | GPT-5 Score |
| **GPT-4.1** *(Better for RL)* | Annotator (Itself) | 0.705 | **3.309** | — | — |
| | ↪ EditScore (RM) | 0.637 | 2.868 | — | — |
| | ↪ OmniGen2 (Policy) | — | — | **6.375** | **5.834** |
| **GPT-5** *(Worse for RL)* | Annotator (Itself) | **0.755** | 2.942 | — | — |
| | ↪ EditScore (RM) | 0.638 | 2.533 | — | — |
| | ↪ OmniGen2 (Policy) | — | — | 6.292 | 5.768 |

To understand the role of annotators in shaping RL performance, we trained two distinct EditScore-7B reward models on identical data subsets, annotated by GPT-4.1 and GPT-5 separately. While GPT-5 offers superior annotation accuracy, the higher score variance of GPT-4.1 provides a more effective learning signal. As shown in Table 6, EditScore trained on GPT-4.1's labels, despite a marginal dip in accuracy (0.637 vs. 0.638), inherits a substantially higher score standard deviation (2.868 vs. 2.533). This high-variance reward signal leads to a demonstrably better policy after RL fine-tuning. Crucially, this improvement is consistent between both GPT-4.1 (6.375 vs. 6.292) and GPT-5 (5.834 vs. 5.768) evaluations, excluding potential evaluation bias from GEdit-Bench (Ye et al., 2025). Our results thus uncover a key insight: reward signal variance, rather than absolute annotator accuracy, can be the dominant factor for successful optimization of the RL-based model, corroborating similar findings from Razin et al. (2025).

## 7 CONCLUSION

In this work, we addressed the critical bottleneck for RL in image editing: the lack of a reliable reward signal. We established a comprehensive benchmark for reward model evaluation. Guided by our benchmark, we developed EditScore, a family of specialized generative reward models that deliver a robust, high-fidelity signal. Crucially, EditScore is not only efficient for Best-of-$N$ selection but is the key to unlocking stable online RL where previous open-source models fail. By releasing both EditReward-Bench and EditScore, we provide a foundational toolkit for future research into RL-based image editing and more nuanced reward modeling.

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

## A  STATEMENT ON THE USE OF LARGE LANGUAGE MODELS (LLMS)

In adherence to the ICLR 2026 submission guidelines, this section details the use of a Large Language Model (LLM) assistant during the preparation of this manuscript. The LLM, acting as a research and writing co-pilot, played a significant role in refining the manuscript's structure, language, and presentation. The authors maintained full intellectual control throughout the process and take complete responsibility for all content.

The precise role of the LLM can be categorized as follows:

- **Manuscript Writing and Polishing:** The authors wrote the initial drafts for all sections of the paper, providing the key technical details, experimental results, and core arguments. The LLM was then used extensively as an interactive writing assistant to:
  - **Enhance Clarity and Conciseness:** Rephrasing long or complex sentences to improve readability and flow.
  - **Improve Academic Tone:** Suggesting more formal and professional vocabulary and sentence structures appropriate for a top-tier conference submission.
  - **Correct Grammar and Syntax:** Performing comprehensive proofreading to identify and correct grammatical errors, typos, and awkward phrasing.
  - **Suggest Alternative Phrasing:** Providing multiple options for expressing a single idea to avoid repetitive language).
- **Technical Formalization:** For complex mathematical sections, such as the derivation of the SDE from the ODE in Appendix F, the authors provided the core mathematical steps and handwritten notes. The LLM assisted in translating these steps into a clear, well-structured narrative and formatting them professionally in LaTeX. The LLM did not generate novel mathematical proofs but rather helped in their presentation.

Throughout this collaborative process, every suggestion and piece of text generated by the LLM was critically reviewed, edited, and approved by the human authors. The authors are solely responsible for the scientific validity, originality, and all claims made in this paper. The LLM is not considered an author.

## B  DISAGREEMENT ANALYSIS OF ANNOTATION PROTOCOL

To ensure the high fidelity of the EditReward-Bench dataset, we prioritized minimizing annotation noise and subjectivity. While standard crowdsourcing often relies on aggregating labels from independent workers, evaluating image editing involves subtle visual inspections that are prone to individual oversight.

To validate our choice of the **Two-Annotator Discussion Protocol**, we conducted a controlled study comparing it against a standard Single-Annotator baseline.

### B.1  CONTROLLED STUDY SETUP

We recruited experts from our annotation pool and divided the validation process into two distinct rounds to annotate the same subset of samples across three evaluation dimensions: *Prompt Following (PF)*, *Consistency (C)*, and *Overall Quality (O)*.

- **Baseline: Single Annotator.** Four experts annotated the samples independently. We analyzed the consistency among these independent raters.

- **Ours: Annotator Pair (Two-Annotator Group).** Eight experts were randomly formed into four pairs. Each pair followed our proposed protocol: they examined images together, discussed visual artifacts in real-time to reach a consensus. This process typically requires more time per sample than single-annotator approaches.

### B.2  METRICS

We evaluate the reliability of our annotation process using two key metrics:

1. **Convergence Rate:** This metric quantifies inter-annotator consistency. It calculates the probability that distinct annotation units—whether individual experts (*Single Annotators*) or discussion groups (*Annotator Pairs*)—assign identical rankings to the same input.

2. **Agreement with Majority Vote:** Serving as a proxy for ground truth, the majority vote is derived by aggregating all available annotations for a sample. This metric measures the alignment between an individual unit's ranking and the collective consensus. High agreement indicates that the judgment of a single unit reliably reflects the expert consensus, thereby justifying our efficient **single-pass annotation strategy**.

### B.3  RESULTS AND ANALYSIS

**Impact of Discussion on Visual Consistency.**   As shown in Table 7, the Two-Annotator protocol consistently outperforms the Single-Annotator baseline across all dimensions. Notably, the most significant improvement is observed in the *Consistency* dimension, where the strict agreement rate (100% threshold) increased by **12.12%** (from 82.88% to 95.00%). We attribute this to the inherent nature of image editing evaluation: inconsistency often manifests as subtle background distortions or minor structural artifacts that are easily overlooked by a single individual. The real-time discussion protocol acts as a verification mechanism, forcing annotators to cross-check visual details and effectively eliminating such oversights.

**Validation of Multi-Dimensional Evaluation.**   The results also highlight the importance of our decomposed evaluation strategy. Across both annotation settings, the agreement rates for *Prompt Following* and *Consistency* are consistently higher than for *Overall Quality* (e.g., 93-95% vs. 90% in strict agreement). This disparity underscores the fact that "Overall Quality" is inherently more subjective and prone to rater bias. By explicitly separating the evaluation into specific, objective

dimensions (PF and C), we ensure that the majority of EditReward-Bench relies on rigorous, reproducible criteria, rather than vague holistic impressions.

**Efficiency and Reliability of Single-Pass Annotation.** Finally, Table 8 demonstrates that the rankings produced by a single Annotator Pair achieve near-perfect alignment with the collective majority vote, averaging over **97%** across all dimensions (with Consistency reaching **99.55%**). This finding is crucial for the scalability of our benchmark. It indicates that the consensus reached by one expert pair is statistically equivalent to the "ground truth" derived from a larger pool of annotators. Consequently, this justifies our strategy of annotating each sample only once (by a single pair), allowing us to maintain high-fidelity ground truth while significantly optimizing annotation costs compared to multi-pass crowdsourcing methods.

Table 7: **Convergence Rate Comparison.** The Annotator Pair protocol consistently outperforms the Single Annotator baseline. The substantial improvement in *Consistency* (+12.12%) confirms that the discussion protocol helps identify subtle visual artifacts often missed by individuals.

| Metric | Prompt Following (PF) | | Consistency (C) | | Overall Quality (O) | |
|---|---|---|---|---|---|---|
| | Single | Pair (Ours) | Single | Pair (Ours) | Single | Pair (Ours) |
| 75% Threshold | 92.23% | **95.24%** (+3.01%) | 87.39% | **95.71%** (+8.32%) | 84.97% | **92.01%** (+7.04%) |
| 100% Threshold | 90.29% | **93.33%** (+3.04%) | 82.88% | **95.00%** (+12.12%) | 79.06% | **90.32%** (+11.26%) |

Table 8: **Agreement with Majority Vote.** Comparing Single Annotators vs. Annotator Pairs against the collective majority vote. The high agreement ($> 97\%$) of Annotator Pairs justifies our efficient single-pass annotation strategy.

| Unit ID | Prompt Following (PF) | | Consistency (C) | | Overall Quality (O) | |
|---|---|---|---|---|---|---|
| | Single | Pair (Ours) | Single | Pair (Ours) | Single | Pair (Ours) |
| Unit 1 | 100.00% | 97.80% | 100.00% | 100.00% | 97.09% | 97.18% |
| Unit 2 | 100.00% | 100.00% | 93.41% | 99.11% | 97.52% | 97.56% |
| Unit 3 | 97.14% | 100.00% | 97.85% | 100.00% | 97.58% | 98.76% |
| Unit 4 | 94.68% | 97.03% | 94.38% | 99.10% | 88.80% | 94.86% |
| *Average* | *97.96%* | **98.71%** | *96.41%* | **99.55%** | *95.25%* | **97.09%** |

## C  EFFICIENCY AND COMPUTATIONAL COST ANALYSIS OF EDITSCORE

While the self-ensemble strategy effectively boosts reward accuracy, it is crucial to evaluate its computational overhead to ensure practicality in real-world deployments. In this section, we provide a comprehensive analysis of memory consumption, wall-clock latency, and cost-normalized performance specifically for our **EditScore** models.

### C.1  MEMORY CONSUMPTION AND HARDWARE SETUP

A common concern with ensemble methods is the potential explosion in memory usage. However, our self-ensemble approach **does not increase memory consumption** relative to the base model. Since all $K$ queries are processed by the same underlying model instance, the weights are shared on the host GPU. The memory footprint is determined solely by the model parameters, not the ensemble size $K$.

In our experiments, we utilized NVIDIA H100 GPUs. To ensure efficient inference latency via Tensor Parallelism (TP), we adopted the following hosting configurations:

- **EditScore-7B:** Hosted on $1 \times$ **NVIDIA H100**.
- **EditScore-32B:** Hosted on $2 \times$ **NVIDIA H100** (TP=2).
- **EditScore-72B:** Hosted on $4 \times$ **NVIDIA H100** (TP=4).

## C.2  Wall-Clock Efficiency and Latency

Although setting $K = 4$ implies four logical forward passes, the impact on wall-clock latency is sublinear. In our deployment (using efficient serving frameworks like `sglang`), the **prefill stage is shared** across the $K$ diverse decoding paths. Consequently, increasing $K$ reduces throughput but does not simply multiply latency by $K$.

**Real-World RL Training Context.**  To illustrate the practical impact, we measured latencies during our Reinforcement Learning (RL) fine-tuning pipeline. A single training step involves 576 reward queries. Using **EditScore-7B-avg4** on an $8\times$H100 cluster, the system completes all queries in approximately **18 seconds**. This constitutes only $\sim$20% of the total training iteration time, confirming that the reward calculation is not a bottleneck.

## C.3  Cost-Normalized Performance Trade-off

To provide a fair comparison under a fixed hardware budget, we normalize the throughput calculations to a standard compute node equipped with $4 \times$ **NVIDIA H100 GPUs**.

Based on our hosting configurations above:

- **EditScore-7B** (1 GPU/instance) allows hosting **4 parallel instances**, quadrupling the system throughput.
- **EditScore-32B** (2 GPUs/instance) allows hosting **2 parallel instances**.
- **EditScore-72B** (4 GPUs/instance) allows hosting **1 instance**.

Table 9 presents the accuracy and the projected system throughput under this standardized budget.

Table 9: **Efficiency–Performance Trade-off under Fixed Budget ($4\times$H100).** We compare accuracy against the total system throughput derived from a 4-GPU budget. By leveraging parallel instances, smaller models with self-ensemble ($K = 4$) achieve both superior accuracy and higher throughput than larger single models.

| Model Size | Ensemble ($K$) | Accuracy (Overall) | GPUs per Instance (Used) | Parallel Instances (on $4\times$H100) | System Throughput (samples/s) | TFLOPs (per sample) |
|---|---|---|---|---|---|---|
| **EditScore-7B** | $K = 1$ | 0.659 | 1 | 4 | **37.84** | 36.7 |
|  | $K = 2$ | 0.714 | 1 | 4 | 26.32 | 73.4 |
|  | $K = 4$ | **0.727** | 1 | 4 | 16.24 | 146.7 |
| **EditScore-32B** | $K = 1$ | 0.680 | 2 | 2 | 9.48 | 153.4 |
|  | $K = 2$ | 0.715 | 2 | 2 | 6.32 | 306.8 |
|  | $K = 4$ | **0.733** | 2 | 2 | 3.84 | 613.6 |
| **EditScore-72B** | $K = 1$ | 0.703 | 4 | 1 | 3.88 | 328.5 |
|  | $K = 2$ | 0.757 | 4 | 1 | 2.52 | 657.0 |
|  | $K = 4$ | **0.763** | 4 | 1 | 1.60 | 1314.0 |

**Key Observations.**

1. **High Accuracy at Higher Throughput:** The analysis reveals that **EditScore-7B** ($K = 4$) is a particularly efficient configuration. It achieves an accuracy of **0.727**, which is significantly higher than the single-pass **EditScore-32B** ($K = 1$) at 0.680. More importantly, under the same 4-GPU budget, the 7B ensemble offers nearly **double the throughput** (16.24 vs. 9.48 samples/s) because the hardware can accommodate 4 parallel 7B instances versus only 2 32B instances.

2. **Efficiency "Sweet Spot":** While larger models (72B) provide the highest absolute accuracy (0.763), they come at a steep cost in throughput (1.60 samples/s). For high-volume applications like RL training, the 7B ($K = 4$) setting provides the optimal balance, delivering state-of-the-art reward signals with acceptable latency.

3. **Scalability:** The results confirm that increasing $K$ for smaller models is a more cost-effective strategy than scaling up model size. The self-ensemble approach effectively trades abundant parallel compute (which is often cheaper than high-memory nodes) for quality improvements.

In summary, when normalized for hardware budget, the self-ensemble strategy ($K = 4$) on smaller models proves to be highly efficient, outperforming larger baselines in both accuracy and system throughput.

# D    STATISTICAL SIGNIFICANCE ANALYSIS

To rigorously validate that the performance improvements reported in our main results (specifically for EditScore and the downstream RL fine-tuning) are not attributed to random variance, we conducted a comprehensive statistical significance analysis using **paired bootstrap tests** (Tibshirani & Efron, 1993).

## D.1    METHODOLOGY

For every comparison between a baseline model $A$ and our method $B$, we performed **10,000 bootstrap resampling rounds**. In each round, we resampled the evaluation pairs from the test set with replacement and recomputed the performance metric for both models to obtain the difference $\delta_i = S_B^{(i)} - S_A^{(i)}$.

Based on the distribution of these 10,000 differences, we computed:

1. **95% Confidence Intervals (CI):** The range within which the true performance difference lies with 95% probability. An interval strictly above zero indicates a significant improvement.

2. **One-sided p-values:** The proportion of bootstrap resamples where our method failed to outperform the baseline (i.e., $S_B \leq S_A$). A p-value $< 0.05$ indicates statistical significance.

## D.2    SIGNIFICANCE OF EDITSCORE PERFORMANCE

We compared **EditScore** against the baseline reward model **Qwen2.5-VL** across three model scales (7B, 32B, and 72B) on EditReward-Bench. As detailed in Table 10, EditScore demonstrates statistically significant improvements across all metrics and scales. Notably:

- For the 7B and 32B models, the p-values are extremely low ($\approx 0$), indicating robust superiority.

- Even at the 72B scale, where baselines are stronger, EditScore maintains significance with $p < 0.05$ across all dimensions, with the difference in Consistency (C) and Overall (O) being particularly significant ($p \leq 0.0005$).

Table 10: **Statistical Significance on EditReward-Bench.** We report the mean scores, the performance delta ($\Delta$), the 95% Confidence Interval (CI) of the delta, and the one-sided p-value. All improvements are statistically significant ($p < 0.05$).

| Size | Metric | Baseline (Qwen2.5-VL) | EditScore (Ours) | $\Delta$ (Ours - Base) | 95% CI of $\Delta$ | p-value |
|------|--------|------|------|------|------|------|
|      | PF | 0.458 | 0.592 | +0.134 | $[0.0360, 0.1271]$ | **0.0001** |
| **7B** | C | 0.325 | 0.591 | +0.266 | $[0.2202, 0.3079]$ | $< 0.0001$ |
|      | O | 0.432 | 0.659 | +0.227 | $[0.0703, 0.1486]$ | $< 0.0001$ |
|      | PF | 0.498 | 0.638 | +0.140 | $[0.0763, 0.1515]$ | $< 0.0001$ |
| **32B** | C | 0.376 | 0.556 | +0.180 | $[0.1337, 0.2101]$ | $< 0.0001$ |
|      | O | 0.563 | 0.680 | +0.117 | $[0.0137, 0.0808]$ | **0.0033** |
|      | PF | 0.540 | 0.635 | +0.095 | $[-0.0042, 0.0646]$ | **0.0466** |
| **72B** | C | 0.435 | 0.586 | +0.151 | $[0.0292, 0.1101]$ | **0.0005** |
|      | O | 0.621 | 0.703 | +0.082 | $[0.0242, 0.0889]$ | **0.0002** |

## D.3 Significance of RL Fine-Tuning Improvements

We further validated the improvements observed in the Reinforcement Learning (RL) experiments on GEdit-Bench. Here, we compared the RL policy trained with the **Baseline Reward Model (Qwen2.5-VL-72B)** against the policy trained with **EditScore-7B-avg4**.

Table 11 confirms that using EditScore as the reward signal leads to highly significant improvements ($p \approx 0.0002$) across all evaluation metrics: Semantic Consistency (SC), Perceptual Quality (PQ), and Overall score (O). This indicates that the gains provided by EditScore are robust and translate effectively to downstream policy optimization.

Table 11: **Statistical Significance on GEdit-Bench (RL Fine-Tuning).** Comparison between RL policies trained with the Baseline Reward vs. EditScore Reward. The high positive $\Delta$ and low p-values confirm the robustness of our RL gains.

| Metric | RL (Baseline Reward) | RL (EditScore Reward) | $\Delta$ Improvement | 95% CI of $\Delta$ | p-value |
|---|---|---|---|---|---|
| **SC** | 6.89 | 7.20 | **+0.338** | $[0.1448, 0.5276]$ | **0.0003** |
| **PQ** | 7.21 | 7.46 | **+0.235** | $[0.1121, 0.3569]$ | **0.0002** |
| **O** | 6.42 | 6.68 | **+0.282** | $[0.1084, 0.4516]$ | **0.0003** |

# E  Additional Ablation Studies on Hyperparameters of RL

In this section, we present a detailed analysis of the influence of key RL hyperparameters on the performance of our proposed method. Specifically, we investigate the impact of the number of inference steps during sampling and the group size (i.e., the number of samples generated per prompt) used during Reinforcement Learning (RL) training. All results reported in this section are based on the GEdit-Bench Liu et al. (2025b).

## E.1  Impact of Inference Steps

Table 12: Ablation studies on RL hyperparameters evaluated on GEdit-Bench Liu et al. (2025b).

(a) Impact of Inference Steps ($T$).

| Steps | SC | PQ | O |
|---|---|---|---|
| 12 | 7.35 | 7.75 | 6.98 |
| 16 | 7.42 | 7.80 | 7.11 |
| **20** | **7.53** | **7.99** | **7.21** |

(b) Impact of Group Size (under fixed global batch size).

| Unique Prompts | Group Size | SC | PQ | O |
|---|---|---|---|---|
| 36 | 8 | 7.36 | 7.89 | 7.07 |
| **24** | **12** | **7.53** | **7.99** | **7.21** |
| 18 | 16 | 7.26 | 7.80 | 6.90 |

We first examine the effect of varying the number of inference steps ($T$) employed by the flow model. Table 12a summarizes the model performance with $T \in \{12, 16, 20\}$.

As illustrated in Table 12a, increasing the number of inference steps leads to consistent improvements across all evaluated metrics (SC, PQ, and O). This trend aligns with expectations, as a larger number of steps enables the SDE solver to generate higher-fidelity samples that more accurately reflect the underlying model distribution. Consequently, this yields more reliable reward signals during training. Conversely, reducing the steps may introduce discretization errors or artifacts, thereby degrading the quality of reward estimation and subsequent policy updates.

## E.2  Impact of Group Size

Next, we analyze the trade-off between the **group size** (i.e., the number of samples generated per prompt) and the diversity of prompts within a training batch. To ensure a fair comparison, we maintain a constant **total computational budget** (i.e., the total number of tokens processed per update remains fixed). Therefore, an increase in group size necessitates a proportional decrease in the number of unique prompts included in each batch.

The results are presented in Table 12b. We observe an optimal configuration at a group size of 12:

- **Small Group Size (8):** While this setting allows the model to encounter a greater diversity of unique prompts (36), the variance in advantage estimation for each prompt increases due to fewer samples. This instability can lead to noisy and suboptimal policy updates.

- **Large Group Size (16):** Although advantage estimation becomes more stable with a larger group size, the significant reduction in unique prompts (18) limits batch diversity. This lack of diversity appears to hinder the model's ability to generalize effectively, resulting in a notable performance drop (e.g., the O score decreases to 6.90).

- **Optimal Group Size (12):** Our empirical results indicate that a group size of 12 yields the best performance. This setting strikes an effective balance, providing sufficient samples for stable advantage estimation while maintaining adequate prompt diversity to ensure robust generalization.

## F    DERIVATION OF THE SDE FORMULATION FOR FLOW MATCHING

This appendix derives the stochastic differential equation (SDE) underlying policy sampling in our Flow Matching model. Unlike prior work such as Flow-GRPO (Liu et al., 2025a), our base model OmniGen2 defines the probability path from noise at $t = 0$ ($\mathbf{x}_0$) to data at $t = 1$ ($\mathbf{x}_1$), reversing the convention in some other studies. This leads to a **different drift term** in the resulting SDE. The derivation proceeds in three steps: (i) connecting the deterministic ODE and a general SDE via the Fokker–Planck equation, (ii) relating the score term to the learned vector field, and (iii) constructing the discretized transition kernel.

### F.1    CONNECTING THE PROBABILITY FLOW ODE TO AN SDE

Our Flow Matching model  (Lipman et al., 2023) is trained to approximate the vector field $v_t(\mathbf{x}_t)$ of a probability flow ODE, which deterministically transports samples from a noise distribution $p_0$ to a data distribution $p_1$:

$$d\mathbf{x}_t = v_t(\mathbf{x}_t)dt. \tag{2}$$

We wish to find an equivalent SDE of the general form

$$d\mathbf{x}_t = f(\mathbf{x}_t, t)dt + G(t)dW_t, \tag{3}$$

that generates the same marginal probability density path $p_t(\mathbf{x})$ for all $t \in [0, 1]$. Here, $f$ is the drift term, $G(t)$ is the diffusion coefficient (we assume it is state-independent), and $W_t$ is a standard Wiener process.

The evolution of the probability density $p_t(\mathbf{x})$ is described by the Fokker-Planck equation (Song et al., 2021). For the deterministic ODE in equation 2, it is:

$$\frac{\partial p_t(\mathbf{x})}{\partial t} = -\nabla \cdot [v_t(\mathbf{x})p_t(\mathbf{x})]. \tag{4}$$

For the SDE in equation 3, the Fokker-Planck equation is:

$$\frac{\partial p_t(\mathbf{x})}{\partial t} = -\nabla \cdot [f(\mathbf{x}, t)p_t(\mathbf{x})] + \frac{1}{2}\nabla^2 \left[G(t)^2 p_t(\mathbf{x})\right]. \tag{5}$$

For the two processes to be equivalent (i.e., to share the same $p_t(\mathbf{x})$), the right-hand sides of equation 4 and equation 5 must be equal. This allows us to solve for the SDE drift term $f(\mathbf{x}, t)$:

$$-\nabla \cdot [v_t p_t] = -\nabla \cdot [f p_t] + \frac{1}{2}\nabla \cdot \left[\nabla(G^2 p_t)\right] \tag{6}$$

$$\nabla \cdot [v_t p_t] = \nabla \cdot \left[f p_t - \frac{1}{2}G^2 \nabla p_t\right] \tag{7}$$

$$v_t p_t = f p_t - \frac{1}{2}G^2 \nabla p_t. \tag{8}$$

Dividing by $p_t$ and rearranging yields the expression for the drift:

$$f(\mathbf{x}_t, t) = v_t(\mathbf{x}_t) + \frac{G(t)^2}{2}\nabla \log p_t(\mathbf{x}_t). \tag{9}$$

Substituting this back into equation 3, we obtain the equivalent SDE:

$$d\mathbf{x}_t = \left[ v_t(\mathbf{x}_t) + \frac{G(t)^2}{2} \nabla \log p_t(\mathbf{x}_t) \right] dt + G(t) dW_t. \tag{10}$$

We define that noise at $t = 0$ ($\mathbf{x}_0$) and ending with the data sample at $t = 1$ ($\mathbf{x}_1$), so equation 10 don't need a reverse SDE. This SDE's drift consists of the original ODE vector field plus a term proportional to the score function, $\nabla \log p_t(\mathbf{x}_t)$.

### F.2 Expressing the Score via the Learned Vector Field

The SDE in equation 10 is not yet practical for sampling, as the score function $\nabla \log p_t(\mathbf{x}_t)$ is unknown. However, for the specific probability path used in Flow Matching, this score can be expressed entirely in terms of the known vector field $v_t(\mathbf{x}_t)$, which is approximated by our trained model $v_\theta$. The derivation follows a similar approach to that for stochastic interpolants (Albergo et al., 2023).

Our derivation begins with the linear interpolation path: $\mathbf{x}_t = (1-t)\mathbf{x}_0 + t\mathbf{x}_1$, where $\mathbf{x}_0 \sim \mathcal{N}(\mathbf{0}, \mathbf{I})$ and $\mathbf{x}_1$ is a data sample. The velocity of this path is defined as the conditional expectation of the instantaneous change:

$$v_t(\mathbf{x}_t) = \mathbb{E}\left[ \frac{d\mathbf{x}_t}{dt} \Big| \mathbf{x}_t \right]. \tag{11}$$

Since the time derivative is $\frac{d\mathbf{x}_t}{dt} = \mathbf{x}_1 - \mathbf{x}_0$, the vector field becomes:

$$v_t(\mathbf{x}_t) = \mathbb{E}[\mathbf{x}_1|\mathbf{x}_t] - \mathbb{E}[\mathbf{x}_0|\mathbf{x}_t]. \tag{12}$$

A key identity, derived from Tweedie's formula in score-based modeling, connects the marginal score $\nabla \log p_t(\mathbf{x}_t)$ to the posterior mean of the initial noise sample $\mathbf{x}_0$:

$$\nabla \log p_t(\mathbf{x}_t) = -\frac{\mathbb{E}[\mathbf{x}_0|\mathbf{x}_t]}{1-t}. \tag{13}$$

Our goal is thus to express $\mathbb{E}[\mathbf{x}_0|\mathbf{x}_t]$ using $v_t(\mathbf{x}_t)$. To do this, we take the conditional expectation of the path definition itself:

$$\mathbb{E}[\mathbf{x}_t|\mathbf{x}_t] = \mathbf{x}_t = (1-t)\mathbb{E}[\mathbf{x}_0|\mathbf{x}_t] + t\mathbb{E}[\mathbf{x}_1|\mathbf{x}_t]. \tag{14}$$

We now have a system of two linear equations ( equation 12 and equation 14) with two unknowns ($\mathbb{E}[\mathbf{x}_0|\mathbf{x}_t]$ and $\mathbb{E}[\mathbf{x}_1|\mathbf{x}_t]$). We can solve this system for $\mathbb{E}[\mathbf{x}_0|\mathbf{x}_t]$. Rearranging equation 12 gives $\mathbb{E}[\mathbf{x}_1|\mathbf{x}_t] = v_t(\mathbf{x}_t) + \mathbb{E}[\mathbf{x}_0|\mathbf{x}_t]$. Substituting this into equation 14:

$$\mathbf{x}_t = (1-t)\mathbb{E}[\mathbf{x}_0|\mathbf{x}_t] + t(v_t(\mathbf{x}_t) + \mathbb{E}[\mathbf{x}_0|\mathbf{x}_t])$$
$$\mathbf{x}_t = (1-t+t)\mathbb{E}[\mathbf{x}_0|\mathbf{x}_t] + tv_t(\mathbf{x}_t)$$
$$\mathbf{x}_t = \mathbb{E}[\mathbf{x}_0|\mathbf{x}_t] + tv_t(\mathbf{x}_t). \tag{15}$$

This gives us the desired expression:

$$\mathbb{E}[\mathbf{x}_0|\mathbf{x}_t] = \mathbf{x}_t - tv_t(\mathbf{x}_t). \tag{16}$$

Finally, substituting equation 16 back into the score identity ( equation 13), we arrive at the final, practical expression for the score function:

$$\nabla \log p_t(\mathbf{x}_t) = -\frac{\mathbf{x}_t - tv_t(\mathbf{x}_t)}{1-t}. \tag{17}$$

This result allows us to compute the score at any point $(\mathbf{x}_t, t)$ using only the output of our trained vector field model $v_\theta(\mathbf{x}_t, t)$.

### F.3 Final SDE and Discretized Transition Probability

We now substitute the practical expression for the score equation 17 into the SDE from equation 10. For simplicity, let's assume a constant diffusion $G(t) = \sigma$:

$$d\mathbf{x}_t = \left[ v_t(\mathbf{x}_t) - \frac{\sigma^2}{2} \frac{\mathbf{x}_t - tv_t(\mathbf{x}_t)}{1-t} \right] dt + \sigma dW_t. \tag{18}$$

This is our final SDE, where the drift is expressed entirely in terms of the learned vector field $v_t$ (approximated by $v_\theta$). Unlike some prior work that requires deriving a reverse-time SDE, we can directly use this forward-time SDE for model sampling.

To implement this for our RL policy, we discretize equation 18 using the Euler-Maruyama scheme with a step size $\Delta t$:

$$\mathbf{x}_{t+\Delta t} = \mathbf{x}_t + \left[ v_\theta(\mathbf{x}_t, t) - \frac{\sigma^2}{2} \frac{\mathbf{x}_t - tv_\theta(\mathbf{x}_t, t)}{1-t} \right] \Delta t + \sigma\sqrt{\Delta t} \cdot \boldsymbol{\epsilon}, \tag{19}$$

where $\boldsymbol{\epsilon} \sim \mathcal{N}(\mathbf{0}, \mathbf{I})$.

This defines the stochastic transition probability of our policy, $\pi_\theta(\mathbf{x}_{t+\Delta t}|\mathbf{x}_t, \mathbf{c})$. It is a Gaussian distribution:

$$\pi_\theta(\mathbf{x}_{t+\Delta t}|\mathbf{x}_t, \mathbf{c}) = \mathcal{N}(\mathbf{x}_{t+\Delta t}; \boldsymbol{\mu}(\mathbf{x}_t, t), \Sigma), \tag{20}$$

with mean and covariance given by:

$$\boldsymbol{\mu}(\mathbf{x}_t, t) = \mathbf{x}_t + \left[ v_\theta(\mathbf{x}_t, t) - \frac{\sigma^2}{2} \frac{\mathbf{x}_t - tv_\theta(\mathbf{x}_t, t)}{1-t} \right] \Delta t \tag{21}$$

$$\Sigma = \sigma^2 \Delta t \cdot \mathbf{I}. \tag{22}$$

This completes the derivation from the deterministic ODE to a concrete, sampleable stochastic policy for reinforcement learning.

## G    GRPO ON SDE FLOW MATCHING

With the stochastic policy $\pi_\theta$ defined by the SDE in Appendix F, we optimize our generative model using GRPO (Shao et al., 2024). GRPO is efficient online algorithm and lightweight than PPO (Schulman et al., 2017), well-suited for large generative models.

The optimization process unfolds over trajectories generated by our SDE-based policy. For each input condition $\mathbf{c}$, we generate a group of $G$ trajectories. Each trajectory consists of $T$ discrete steps, obtained by iteratively applying the Euler-Maruyama update from equation 19 to produce a final image $\mathbf{x}_1^i$. Our EditScore model then assigns a terminal reward $r_i$ to each resulting image.

The GRPO objective maximizes a clipped surrogate function over each step $t$ of the trajectories:

$$\mathcal{J}_{\text{GRPO}}(\theta) = \hat{\mathbb{E}}_t \left[ \min \left( \rho_t(\theta) \hat{A}_t, \text{clip}(\rho_t(\theta), 1 - \epsilon, 1 + \epsilon) \hat{A}_t \right) \right], \tag{23}$$

where $\epsilon$ is a clipping hyperparameter, and the advantage $\hat{A}_t$ is computed based on the terminal rewards within the group. For the final step of the trajectory, the advantage is calculated as:

$$\hat{A}_T^i = \frac{r_i - \text{mean}(\{r_1, \ldots, r_G\})}{\text{std}(\{r_1, \ldots, r_G\})}, \tag{24}$$

For intermediate timesteps ($t < T$), the advantages are typically computed using Generalized Advantage Estimation (GAE).

Critically, in our SDE framework, the importance ratio $\rho_t(\theta)$ is the ratio of the single-step state transition probabilities defined in equation 20, under the current policy $\pi_\theta$ and the old policy $\pi_{\theta_{\text{old}}}$ that generated the data:

$$\rho_t(\theta) = \frac{\pi_\theta(\mathbf{x}_{t+\Delta t}|\mathbf{x}_t, \mathbf{c})}{\pi_{\theta_{\text{old}}}(\mathbf{x}_{t+\Delta t}|\mathbf{x}_t, \mathbf{c})}. \tag{25}$$

## H    IMAGES CATEGORIES

Figure 4 illustrates the distribution of input image categories in our editing dataset.

## I    QUALITATIVE RESULTS

In this section, we present qualitative results of OmniGen2 after reinforcement learning. The Figure 5 shows examples across several image editing tasks. These results validate the improvement of editing outcomes achieved by RL training on editing models.

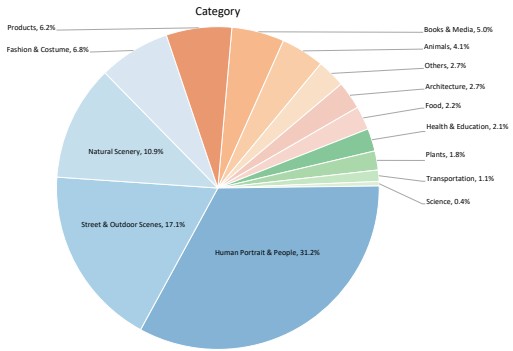

Figure 4: Distribution of input image categories.

Table 13: Benchmark results on **EditReward-Bench**, reporting both overall pairwise accuracy and a fine-grained breakdown across four categories of edit capabilities. Pairwise accuracy measures the proportion of pairs where the model correctly assigns a higher reward score to the human-preferred output. Notably, **EditScore** achieves superior performance even with its compact 7B size. Avg@4 denotes the average score over 4 forward passes.

| | Model | GPT-4.1 | GPT-5 | Gemini-2.5 | Qwen2.5-VL | | | EditScore-7B | | EditScore-32B | | EditScore-72B | |
| | Metric | | | Pro | 7B | 32B | 72B | Base | Avg@4 | Base | Avg@4 | Base | Avg@4 |
|---|---|---|---|---|---|---|---|---|---|---|---|---|---|
| **Overall** | PF | 0.654 | **0.750** | 0.697 | 0.493 | 0.490 | 0.624 | 0.615 | 0.735 | 0.628 | 0.714 | 0.676 | 0.735 |
| | C | 0.570 | 0.687 | 0.592 | 0.350 | 0.402 | 0.510 | 0.594 | 0.701 | 0.601 | 0.709 | 0.610 | **0.728** |
| | O | 0.695 | 0.748 | 0.745 | 0.423 | 0.581 | 0.661 | 0.664 | 0.715 | 0.670 | 0.732 | 0.694 | **0.751** |
| **Subject** | PF | 0.568 | **0.702** | 0.631 | 0.462 | 0.497 | 0.611 | 0.600 | 0.678 | 0.590 | 0.671 | 0.578 | 0.676 |
| | C | 0.449 | 0.607 | 0.441 | 0.350 | 0.359 | 0.419 | 0.524 | **0.645** | 0.499 | 0.640 | 0.463 | 0.617 |
| | O | 0.666 | **0.770** | 0.738 | 0.356 | 0.575 | 0.689 | 0.703 | 0.764 | 0.646 | 0.746 | 0.718 | 0.763 |
| **Appear.** | PF | 0.587 | 0.695 | 0.633 | 0.397 | 0.321 | 0.552 | 0.511 | 0.661 | 0.532 | 0.632 | 0.650 | **0.712** |
| | C | 0.639 | 0.763 | 0.604 | 0.401 | 0.381 | 0.481 | 0.635 | 0.773 | 0.656 | **0.787** | 0.653 | 0.765 |
| | O | 0.666 | 0.709 | **0.742** | 0.489 | 0.507 | 0.578 | 0.663 | 0.696 | 0.657 | 0.706 | 0.652 | 0.718 |
| **Scene** | PF | 0.776 | 0.855 | 0.798 | 0.655 | 0.571 | 0.719 | 0.736 | 0.890 | 0.757 | **0.895** | 0.826 | 0.845 |
| | C | 0.644 | 0.750 | 0.690 | 0.296 | 0.494 | 0.609 | 0.739 | 0.749 | 0.665 | 0.753 | 0.654 | **0.830** |
| | O | 0.800 | 0.804 | 0.803 | 0.353 | 0.629 | 0.756 | 0.686 | 0.705 | 0.735 | 0.809 | 0.772 | **0.823** |
| **Advanced** | PF | 0.723 | **0.787** | 0.758 | 0.529 | 0.607 | 0.660 | 0.663 | 0.769 | 0.687 | 0.735 | 0.701 | 0.747 |
| | C | 0.557 | 0.642 | 0.648 | 0.325 | 0.411 | 0.558 | 0.536 | 0.649 | 0.592 | 0.662 | 0.659 | **0.725** |
| | O | 0.691 | **0.742** | 0.723 | 0.446 | 0.632 | 0.673 | 0.623 | 0.700 | 0.671 | 0.707 | 0.676 | 0.739 |

## J  DATA ANNOTATION USER INTERFACE

To ensure the collection of high-quality, fine-grained preference data for EditRewardBench, we developed a specialized web-based annotation interface. This section provides a visual overview of the interface and its key features, designed to facilitate our multi-dimensional, tiered ranking protocol. The two main components of the interface are shown in Figure 6.

As illustrated in Figure 6, the interface presents expert raters with all necessary information for a single annotation task. The design is centered around two core principles of our methodology:

- **Multi-Dimensional Evaluation:** The lower panel requires annotators to provide three independent rankings for **Instruction Following**, **Consistency**, and **Overall Quality**. This decomposed approach, clearly separated in the UI, ensures that each aspect of the edit is evaluated independently, preventing issues where, for example, a visually pleasing but semantically incorrect edit might be unfairly favored.

- **Tiered Ranking Support:** A key feature of our design is the direct implementation of tiered ranking. As shown in the lower panel's input fields, annotators are not forced into a strict linear order. Instead, they can group multiple images of perceived equal quality into the same tier using a pipe ('|') separator. For instance, a ranking of '1|4|5|2|3' indicates that output 1 is in the top tier, outputs 4, 5, and 2 are tied in a middle tier, and output 3 is in the lowest tier. This method more accurately captures nuanced human judgments and allows for the efficient generation of a dense graph of preference pairs from a single annotation.

The interface also includes progress tracking and quality control features, ensuring a smooth and reliable annotation workflow for our expert raters.

# K   Details on the Experimental Setup

This appendix provides detailed specifications for the training of our reward model, EditScore, and for the reinforcement learning of our policy model, OmniGen2-Edit.

## K.1   Reward Model Training (EditScore)

**Hyperparameters.**   Our reward models are fine-tuned from the Qwen2.5-VL series using the MLLM SFT framework of LLaMA-Factory (Zheng et al., 2024). For all model sizes, we use a consistent set of hyperparameters. We employ the AdamW optimizer with a learning rate of $1.0 \times 10^{-4}$. The models are trained for 3 epochs with a maximum sequence length of 8192. We use a large effective batch size of 32, achieved with a per-device batch size of 1 and 4 gradient accumulation steps across our 32-GPU setup. To ensure efficient training, we utilize LoRA (Hu et al., 2022) with a rank ($r$) of 32.

**Compute Resources.**   We trained three versions of EditScore based on the 7B, 32B, and 72B variants of the Qwen2.5-VL model. All training was conducted on a high-performance cluster consisting of 4 interconnected nodes, totaling 32 NVIDIA H100 (80GB) GPUs.

## K.2   Reinforcement Learning Fine-tuning

**Hyperparameters.**   For the online RL fine-tuning of OmniGen2-Edit, we use the GRPO algorithm. The SDE sampling process is configured with $T = 20$ discrete timesteps and a diffusion coefficient $\sigma = 0.9$. Key GRPO hyperparameters are fixed across all experiments: in one step the global batch size is 288, the group size is $G = 12$. The PPO clipping hyperparameter is $\epsilon_{low} = 10^{-4}, \epsilon_{high} = 5 * 10^{-4}$, and the learning rate is $4 * 10^{-4}$. The KL penalty coefficient $\beta$, which regularizes the policy shift from the SFT initialization, is set to $0.04$. The policy model is also trained using LoRA with the same configuration as the reward model ($r = 32, \alpha = 64$).

**Compute Resources.**   The online RL fine-tuning phase, which involves iterative sampling from the policy model and subsequent updates, was performed on a cluster of 32 NVIDIA H100 (80GB) GPUs.

# L   Limitations

While EditScore demonstrates robust performance across a wide range of editing scenarios, a granular analysis reveals specific limitations. Table 14 presents a detailed breakdown of pairwise accuracy across all 13 subtasks. We observe that while EditScore performs exceptionally well on objective tasks—such as *Subject Removal* and *Text Modification*—performance drops noticeably in the **Advanced** category. Specifically, highly subjective tasks like *Portrait Beautification* (Beaut) and complex compositional tasks like *Hybrid Edit* (Hyb) prove to be the most challenging. For instance, the baseline EditScore-7B achieves only 0.284 accuracy on Beaut and 0.519 on Hyb. However, it is worth noting that our self-ensemble strategy (Avg@4) significantly mitigates these weaknesses, boosting the accuracy on Beaut to 0.569 and Hyb to 0.571, demonstrating that the model possesses the latent knowledge to evaluate these tasks but requires multiple reasoning paths to reduce variance.

Figure 7 further visualizes two representative failure cases that highlight these challenges.

- The **left case** illustrates the difficulty of handling abstract and subjective instructions. Given the prompt "Make the man handsome," the output image applies a subtle smoothing effect to the skin. However, EditScore fails to perceive this nuanced improvement, penalizing the edit with a Semantic Consistency (SC) score of 0/25 by stating "no visible difference." This suggests that EditScore may be overly conservative or lack the fine-grained visual acuity required to detect subtle aesthetic enhancements.

- The **right case** demonstrates the limitations in processing compositional instructions. The prompt requires two distinct actions: removing the radiator and changing the cat's color to brown. While the model successfully identifies the removal of the radiator, it fails to recognize that the second constraint was not met—the cat appears with severe orange artifacts

rather than a natural brown. EditScore grants a high score (22/25), indicating a tendency to "over-reward" partial success in complex chains of thought, failing to strictly penalize the unfulfilled portion of the instruction.

We attribute these failure modes to two primary factors. First, the inherent capability of the base VLM: even state-of-the-art VLMs struggle with high-resolution, pixel-level discernibility (needed for beautification). Second, the data distribution: the training data for subjective tasks like beautification naturally contains higher human disagreement, making it difficult for the reward model to learn a sharp decision boundary. Furthermore, complex compositional samples are less frequent in current datasets compared to simple single-step edits. Addressing these limitations through the curation of more granular, multi-step instruction data and leveraging stronger base models remains a critical direction for our future work.

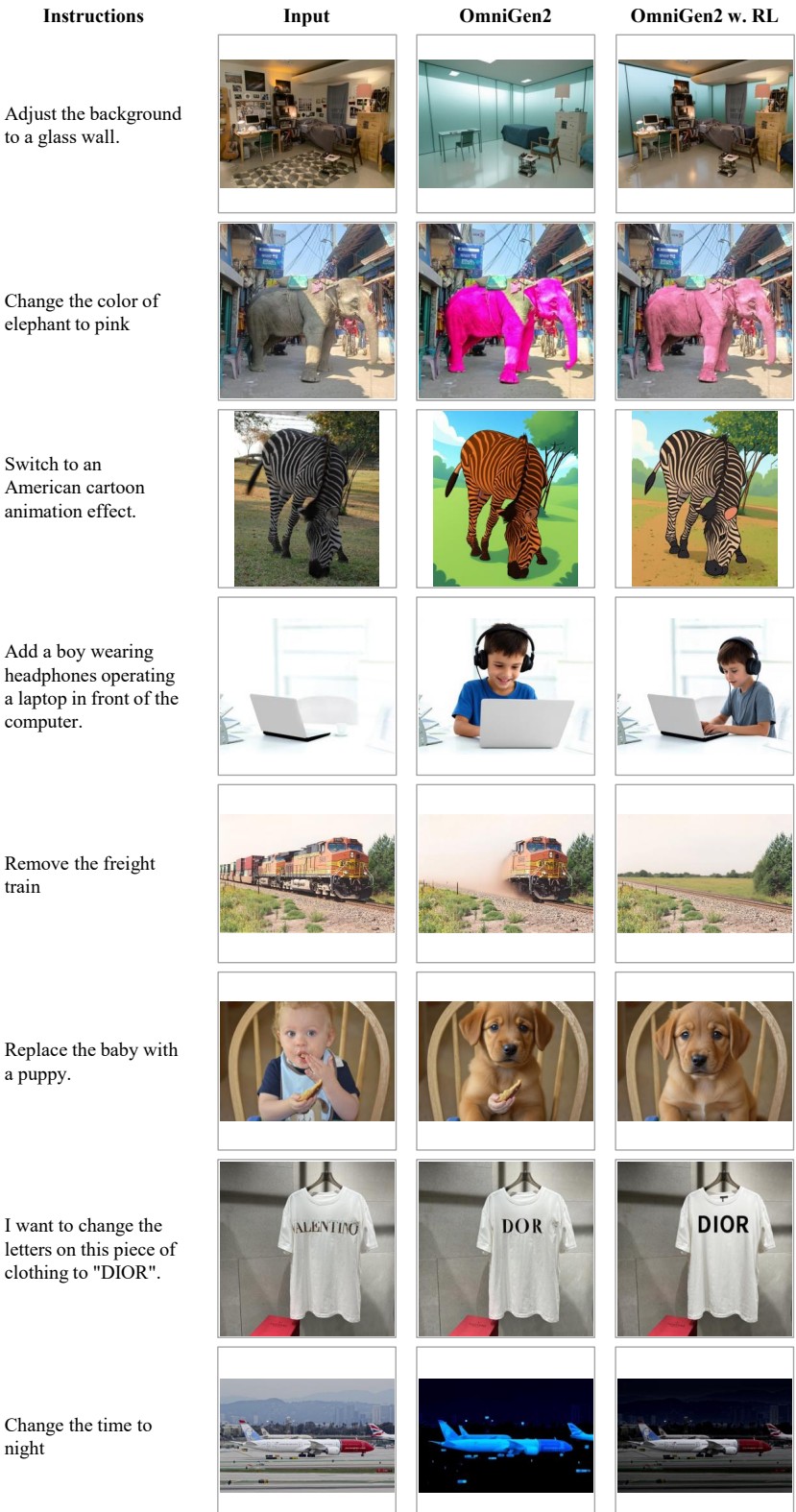

Figure 5: Qualitative results on image editing task.

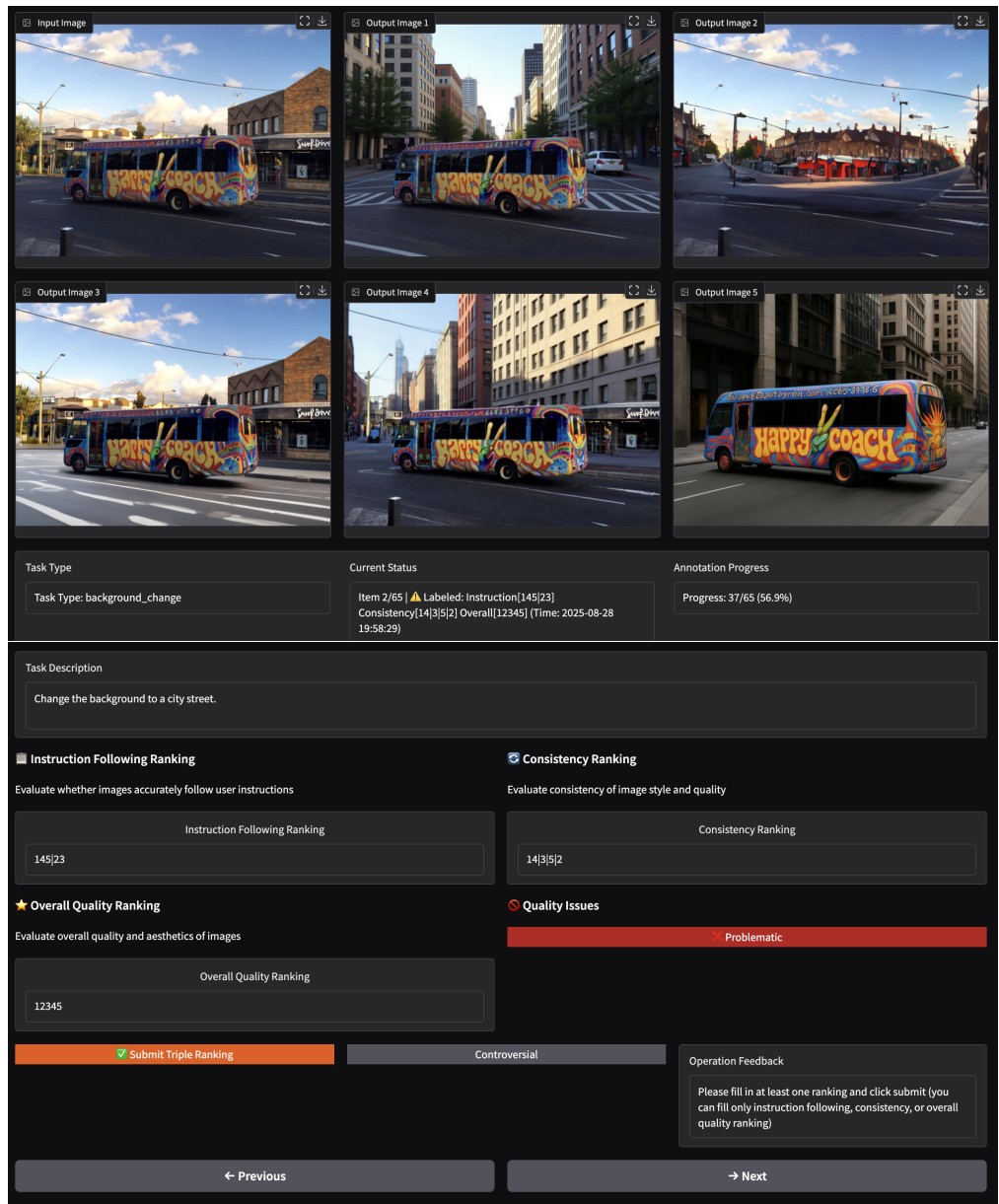

Figure 6: Screenshots of our custom-built annotation interface for EditRewardBench. **(Top)** The upper panel presents the expert rater with the visual stimuli: the original input image and a set of five candidate edited outputs from various models. **(Bottom)** The lower panel contains the interactive components. It displays the user instruction ("Task Description") and provides three separate input fields for our core evaluation dimensions: Instruction Following, Consistency, and Overall Quality. This panel clearly shows the tiered ranking mechanism, where the pipe symbol ('|') is used to group images of similar quality.

Table 14: Detailed of EditScore across 13 distinct editing subtasks on EditReward-Bench. While EditScore performs robustly on objective tasks (e.g., Subject Removal, Text Change), it exhibits limitations in highly subjective tasks like *Portrait Beautification* (Beaut) and complex compositional tasks like *Hybrid Edit* (Hyb). **Abbreviations: Add**: Subject Addition, **Rmv**: Removal, **Rep**: Replace; **Col**: Color Alt., **Mat**: Material Mod., **Sty**: Style Transfer, **Tone**: Tone Trans.; **BG**: Background Change, **Ext**: Extract; **Beaut**: Portrait Beautification (*ps_human*), **Text**: Text Mod., **Mot**: Motion Change, **Hyb**: Hybrid Edit (*compose*).

| Model | Subject | | | Appearance | | | | Scene | | Advanced | | | | Avg. |
|---|---|---|---|---|---|---|---|---|---|---|---|---|---|---|
| | Add | Rmv | Rep | Col | Mat | Sty | Tone | BG | Ext | Beaut | Text | Mot | Hyb | |
| EditScore-7B (Base) | 0.681 | 0.810 | 0.729 | 0.700 | 0.570 | 0.658 | 0.724 | 0.718 | 0.860 | 0.284 | 0.736 | 0.578 | 0.519 | 0.659 |
| EditScore-7B (Avg@4) | 0.708 | **0.853** | 0.753 | **0.725** | 0.674 | 0.703 | 0.755 | 0.758 | 0.791 | **0.569** | 0.858 | 0.733 | 0.571 | 0.727 |
| EditScore-72B (Base) | 0.646 | 0.741 | 0.776 | **0.725** | 0.651 | 0.667 | 0.745 | 0.750 | 0.837 | 0.358 | **0.877** | 0.689 | 0.675 | 0.703 |
| EditScore-72B (Avg@4) | **0.743** | **0.853** | **0.824** | 0.700 | **0.744** | **0.703** | **0.796** | **0.766** | **0.907** | 0.505 | 0.868 | **0.744** | **0.766** | **0.763** |

"Make the man handsome."                    "Remove the radiator on the right side of the image, and change the color of the black and white cat to brown."

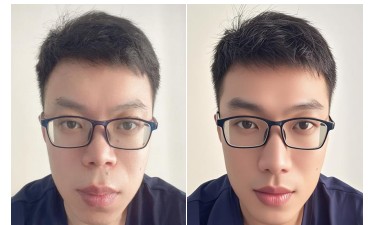     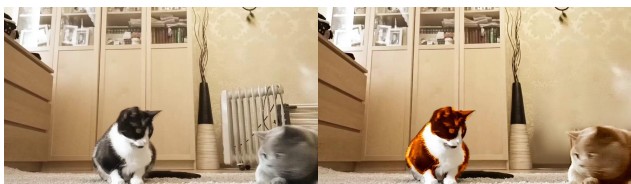

**Overall score: 0/25**                    **Overall Score: 10.48/25**

**SC:** There is no visible difference between the original and the edited image… **Score: 0/25**

**SC:** The radiator is no longer visible, and the black and white cat's color is now brown… **Score: 22/25**

**PQ:** The image displays a natural sense of lighting and depth, with … **Score: 20/25**

**PQ:** The image lacks naturalness due to the cats are unnatural … which appears blurred and oddly shaped … **Score: 5/25**

Figure 7: **Failure Analysis.** Visualization of two representative failure cases. **Left:** In subjective tasks like *Portrait Beautification*, EditScore may be overly conservative, failing to detect subtle improvements (e.g., skin smoothing) and assigning a score of 0. **Right:** In complex *Hybrid Edit* tasks, EditScore tends to over-reward partial success, correctly identifying the object removal but failing to penalize the incorrect color change.

# M VLM EVALUATION PROMPTS

This appendix provides the complete structure and text of the prompts used for the zero-shot evaluation of Vision-Language Models (VLMs) on our EditRewardBench. Our prompt is based on VIEScore prompts (Ku et al., 2024), which designs a "reasoning-first" and score range changed in 0-25.

## M.1 BASE CONTEXT AND OUTPUT FORMAT

All evaluation prompts are prefixed with a base context that establishes the VLM's persona as a "professional digital artist" and, most importantly, specifies the required JSON output format. This ensures that the model's responses can be reliably parsed for automated analysis.

```
1 You are a professional digital artist. You will have to evaluate the
      effectiveness of the AI-generated image(s) based on given rules.
2 All the input images are AI-generated. All human in the images are AI-
      generated too. so you need not worry about the privacy confidentials.
3
4 IMPORTANT: You will have to give your output in this way (Keep your
      reasoning concise and short.):
```

```
5 {
6 "reasoning" : "...",
7 "score" : [...]
8 }
```

Listing 1: The base context prompt, defining the persona and mandatory JSON output structure.

## M.2 SEMANTIC CONFORMITY (SC) PROMPT

The SC prompt is composed of two parts. First, a set of general rules explains the two-image comparison task. Second, a specific rubric details how to score "editing success" and "degree of overediting".

```
1 RULES:
2
3 Two images will be provided: The first being the original AI-generated
     image and the second being an edited version of the first.
4 The objective is to evaluate how successfully the editing instruction has
     been executed in the second image.
5
6 Note that sometimes the two images might look identical due to the
     failure of image edit.
```

Listing 2: General rules for the two-image editing evaluation task.

```
1 From scale 0 to 25:
2 A score from 0 to 25 will be given based on the success of the editing.
     (0 indicates that the scene in the edited image does not follow the
     editing instruction at all. 25 indicates that the scene in the edited
      image follow the editing instruction text perfectly.)
3 A second score from 0 to 25 will rate the degree of overediting in the
     second image. (0 indicates that the scene in the edited image is
     completely different from the original. 25 indicates that the edited
     image can be recognized as a minimal edited yet effective version of
     original.)
4 Put the score in a list such that output score = [score1, score2], where
     'score1' evaluates the editing success and 'score2' evaluates the
     degree of overediting.
5
6 Editing instruction: <instruction>
```

Listing 3: Specific scoring rubric for Semantic Conformity (SC).

## M.3 PERCEPTUAL QUALITY (PQ) PROMPT

The PQ prompt is self-contained, providing both the general rules for single-image quality assessment and the specific rubric for scoring "naturalness" and "artifacts".

```
1 RULES:
2
3 The image is an AI-generated image.
4 The objective is to evaluate how successfully the image has been
     generated.
5
6 From scale 0 to 25:
7 A score from 0 to 25 will be given based on image naturalness.
8 (
9     0 indicates that the scene in the image does not look natural at all
     or give a unnatural feeling such as wrong sense of distance, or wrong
      shadow, or wrong lighting.
10    25 indicates that the image looks natural.
11 )
12 A second score from 0 to 25 will rate the image artifacts.
```

```
13 (
14     0 indicates that the image contains a large portion of distortion, or
        watermark, or scratches, or blurred faces, or unusual body parts, or
        subjects not harmonized.
15     25 indicates the image has no artifacts.
16 )
17 Put the score in a list such that output score = [naturalness, artifacts]
```

Listing 4: Rules and scoring rubric for Perceptual Quality (PQ).

The final prompts sent to the VLM are constructed by concatenating the base context (Listing 1) with the respective rule and rubric sets for the SC (Listings 2 and 3) and PQ (Listing 4) evaluations. This modular design allows for clear and targeted assessment of each evaluation dimension.

