# OpenReview forum: "EditScore: Unlocking Online RL for Image Editing via High-Fidelity Reward Modeling"
_ICLR.cc/2026/Conference — ICLR 2026 Poster_

### Official Review · Reviewer_tH54 · 2025-10-28

**Soundness:** 2
**Presentation:** 2
**Contribution:** 4
**Rating:** 4
**Confidence:** 3

**Summary:**

This paper introduces EditReward-Bench (a benchmark for reward models) and EditScore (a family of generative reward models) for image editing. The paper first collects human preference and ranking data from annotators based on candidate images and asked to rank them based on certain qualities. This data is then used to train generative reward models, which are compared for agreeableness to the human rankers.

**Strengths:**

- this paper introduces a solid dataset and reward model for RL training.
- this paper is decently written, but some sections (eg, explaining how the scores of table 2 of EditReward-Bench) could be rewritten. Otherwise, it's clear.
- the methodology for data collection seems sound.

**Weaknesses:**

- I am concerned about the use of candidate images from current models as part of the data labeling process. Indeed, if there is an area where all the candiate models struggle, although we would be able to provide a relative ranking, the maximal reward would only be obtained from the model generalizing, not any supervision.
- section 6.4.1 compares two different models for optimization, and as OmineGen2 was chosen for high pass@k, a better comparison would be using the same base model. It is hard to tell if this truly is the result of a better reward model, or a more amenable base model.

**Questions:**

- how are table 2 scores computed exactly?
- how are the aggregation scores computed?

---

> ### Author Response · Authors · 2025-11-24
> **Official Response to Reviewer tH54 (Part 1 of 2)**
>
> We appreciate your meticulous feedback and we are delighted by your positive remarks on the clarity of the writing and the soundness of our approach. Below, we provide a detailed response:
>
> > **W1: I am concerned about the use of candidate images from current models as part of the data labeling process. Indeed, if there is an area where all the candidate models struggle, although we would be able to provide a relative ranking, the maximal reward would only be obtained from the model generalizing, not any supervision.**
>
> **A1**: We thank the reviewer for raising this important concern. We fully agree that relying on candidate outputs from existing editing models may create cases where all models fail on certain instructions, making the collected annotations only provide relative supervision.
>
> To address this, we considered two aspects. First, we curate the candidate pool using the most advanced image-editing models available, such as NanoBanana, to ensure a sufficiently high upper bound on candidate quality. Second, as the reviewer noted, achieving the maximal reward may require model generalization, which we view as an intended and beneficial property. The goal of the reward model is not to mimic the candidate models but to learn human editing preferences, thereby encouraging the generation of outputs that surpass all candidates.
>
>
> > **W2: section 6.4.1 compares two different models for optimization, and as OmineGen2 was chosen for high pass@k, a better comparison would be using the same base model. It is hard to tell if this truly is the result of a better reward model, or a more amenable base model.**
>
> **A2:** We appreciate the reviewer’s observation and agree that comparing different base models can confound the interpretation in Section 6.4.1.
>
> To directly control for this factor, we conducted **new experiments using the *same* base model (OmniGen2)** while varying only the reward model (GPT-4.1, EditScore-7B, and EditScore-qwen3-vl-8B). We further included Qwen2.5-VL-7B/72B reward baselines for completeness. EditScore-qwen3-vl-8B is trained on the newly released Qwen3-VL-8B backbone using the same dataset.
>
> The results are summarized below:
>
> | Model                           | GEdit (SC) | GEdit (PQ) | GEdit (O) | ImgEdit (O) | EmuEdit (CLIP-I) | EmuEdit(CLIP-Out) | EmuEdit(DINO) |
> | ------------------------------- | :--------: | :--------: | :-------: | :---------: | :--------------: | :---------------: | :-----------: |
> | **(Group: GEdit-Bench-EN)**     |            |            |           |             |                  |                   |               |
> | OmniGen2 (no RL)                |    6.72    |    7.20    |    6.28   |     3.40    |       0.876      |       0.309       |     0.822     |
> | RL w/ Qwen2.5-VL-7B             |    6.87    |    6.86    |    6.22   |     N/A     |         —        |         —         |       —       |
> | RL w/ Qwen2.5-VL-72B            |    6.89    |    7.21    |    6.42   |     3.60    |       0.871      |       0.311       |     0.814     |
> | RL w/ **GPT-4.1**               |   *7.24*   |    7.41    |   *6.73*  |   **3.66**  |       0.884      |     **0.312**     |     0.843     |
> | **RL w/ EditScore-7B**          |    7.20    |   *7.46*   |    6.68   |    *3.63*   |     **0.900**    |      *0.310*      |   **0.870**   |
> | **RL w/ EditScore-qwen3-vl-8B** |  **7.27**  |  **7.77**  |  **6.89** |     3.62    |      *0.891*     |      *0.310*      |    *0.861*    |
>
> These controlled experiments show that:
>
> * Even when trained on the **same base model**, EditScore consistently outperforms the GPT-4.1 reward model on **most metrics** across GEdit-Bench-EN, ImgEdit, and EmuEdit.
> * EditScore-qwen3-vl-8B further delivers the **strongest overall performance**, suggesting that EditScore’s improvements generalize across reward model architectures.
>
> Therefore, the performance gains observed in Section 6.4.1 stem from a **better reward model**, not from differences in the base model.
>
> > **Q1: how are table 2 scores computed exactly?**
>
> **A3**: Thank you for raising this point. The scores in Table 2 represent the preference prediction accuracy of the reward models against the established human preference.  Specifically, the metric is calculated by determining the proportion of the candidate pairs $(A, B)$ for which the reward model assigns a higher score to the human-preferred output $A$, i.e., $S(A) > S(B)$.
>
> We have add this clarification to the caption of Table 2 and provide more detailed explanation in Section 3.

---

> ### Author Response · Authors · 2025-11-24
> **Official Response to Reviewer tH54 (Part 2 of 2)**
>
> > **Q2: how are the aggregation scores computed?**
>
> **A4**: Thanks for your question. We recognize that this detail was not sufficiently clear in the original manuscript.
>
> The final aggregation scores $S_{final}$ are computed using an arithmetic mean across K independent passes scores $\{s_{1}, s_{2},..., s_{K}\}$ :
> $$
> S_{final} = \frac{1}{K} \sum_{i=1}^{K}s_{i}
> $$
> We have revised Section 4.1.2 (Inference-Time Ensembling Startegy) of the manuscript to include this formula and a detailed explanation of the calculation process, which we believe fully addresses your concern.

---

> > ### Comment · Reviewer_tH54 · 2025-11-26
> >
> > I thank the authors for their response. I am mostly satisfied with my concerns (especially regarding the added RL experiments). Further, I am convinced as to the practical utility of the reward model as an RL objective.
> >
> > However, I maintain my core concern is regarding the data collection methodology. Indeed, the models that are used to collect preference pairs are state-of-the-art, but that does not mean they are without failure modes or low quality generations.  While I agree that the goal is to align to human preferences, many of the prompts seem to be assessing the ability of the model to follow instructions. Why in this case, would it not be better than simply use a real image as the ground truth and a generated image as a negative sample or some other technique which would more more attuned to the prompts?

---

> > > ### Author Response · Authors · 2025-11-27
> > > **Response to Remaining Concern on Data Collection Methodology (Part 1 of 2)**
> > >
> > > We sincerely thank the reviewer for the thoughtful follow-up and for acknowledging the value of the newly added RL experiments. We understand that your *core* concern lies in the data collection methodology, and we address it below with additional clarification and empirical evidence.
> > >
> > > ### **(1) Why relative comparisons among generated edits—rather than real images—are the right supervision for *instruction-based image editing***
> > >
> > > We fully agree that using real images as ground-truth positives is appealing for reconstruction-style tasks.
> > > However, instruction-based image editing has a very different goal:
> > >
> > > > **The objective is not to reproduce a “correct image”, but to produce an image that best satisfies the user’s editing intent.**
> > >
> > > A real image rarely corresponds to a uniquely correct outcome for an edit instruction. For example:
> > >
> > > * “make the dog happier”
> > > * “turn this dress into a winter coat”
> > > * “replace the background with a futuristic city at night”
> > >
> > > These do **not** have a single real image that represents the “true” target.
> > > Using a single real image as the positive label enforces:
> > >
> > > * a **single canonical solution**,
> > > * penalizes many equally valid edits, and
> > > * forces the reward to learn *photorealism similarity*, not *instruction satisfaction or human preference*.
> > >
> > > This becomes particularly problematic when the user instruction describes *counterfactual*, *future*, *stylized*, or *multi-step* edits, where ground-truth images simply do not exist.
> > >
> > > This is why methods such as MagicBrush, OmniGen, OmniGen2, Echo4o—all state-of-the-art image editing models—are trained on **synthetic comparison data**, not on real-image targets.
> > >
> > > Our reward model aims to capture this notion of “which edited result better aligns with human intent”, rather than “which image is closest to a fixed ground-truth”.
> > >
> > > ### **(2) Why our synthetic comparisons are not “relative only among bad models”**
> > >
> > > To directly address your concern about candidate quality, we performed a *small-scale user study* over the preference groups used in the benchmark.
> > > Four independent annotators evaluated whether each preference group contains at least one result that can be considered a *near-ground-truth* interpretation of the instruction.
> > >
> > > > **They judged that ~95% of groups contain at least one such high-quality candidate.**
> > >
> > > This indicates that—even though models are not perfect—the candidate pool reliably contains high-quality edits.
> > >
> > > More importantly, the **quality diversity** is intentional and beneficial:
> > >
> > > * we deliberately sample from **11 models**, ranging from weaker models to the strongest available (NanoBanana)
> > > * this produces **fine-grained variation** along realism, instruction-following, region preservation, and aesthetics
> > > * human preference labels across such diverse outputs allow the reward to capture nuanced distinctions that cannot be learned from a single real-image target.
> > >
> > > ### **(3) Why “real image as GT + generated as negative” is not sufficient or scalable**
> > >
> > > We respectfully note that adopting this paradigm would require access to real edited pairs:
> > >
> > > > (input image, desired edited output)
> > >
> > > Such data is extremely difficult to collect at scale, as demonstrated by previous attempts such as UniReal and OmniGen2’s video-based pipeline:
> > >
> > > * long-term temporal tracking fails frequently
> > > * many edit categories do not occur naturally in real videos
> > > * backgrounds and identities drift, invalidating the pair
> > > * only a very narrow set of “accidental real edits” can be captured through videos
> > >
> > > Hence, real-pair-based methods cannot cover the broad editing categories used in practice, such as:
> > > style change, material change, global relighting, fantasy edits, replacing objects with novel concepts, etc.
> > >
> > > Synthetic relative comparisons remain the **only scalable and broadly applicable** supervision for general-purpose image editing RL.
> > >
> > > ### **(4) On “prompts assessing instruction-following ability”**
> > >
> > > We would like to clarify that instruction-following is **inherently a core dimension of real-world editing preference**, rather than an artifact of our prompt design:
> > >
> > > * In practical editing workflows, users overwhelmingly prioritize *“Did the model follow my instruction?”* before considering aesthetic refinements or subtle perceptual differences.
> > > * Consequently, instruction-following is itself an essential component of human editing preference, not a confounding factor.
> > >
> > > At the same time, our evaluation explicitly incorporates **multiple dimensions**. The *“overall quality”* dimension captures subtle aspects such as aesthetic preference. Our disagreement analysis (**please ref to A1 of our response to Reviewer bWV1**) further shows that annotators exhibit **higher variance in “overall quality”** than in “instruction following,” indicating that we do not neglect nuanced human preference—rather, we evaluate it through a separate, dedicated dimension.

---

> > > ### Author Response · Authors · 2025-11-27
> > > **Response to Remaining Concern on Data Collection Methodology (Part 2 of 2)**
> > >
> > > ### **(5) Acknowledging the limitation and moving forward**
> > >
> > > We appreciate the reviewer’s perspective, and we acknowledge that incorporating real-image targets—when available—could complement our dataset.
> > > As a direct follow-up, we are adding a small subset of real edited pairs from the X2I2 dataset of OmniGen2 into the next version of our reward model training data and benchmark as an **auxiliary signal**, while maintaining synthetic preference comparisons as the primary source.
> > >
> > > We believe this hybrid approach will further strengthen the reward model, without losing the diversity and instruction coverage that current real pairs cannot provide.
> > >
> > > **We hope these clarifications fully resolve your remaining concern. If you feel that our response and the additional experiments have strengthened the paper, we would greatly appreciate it if you could reconsider your score.**

---

> > > > ### Comment · Reviewer_tH54 · 2025-11-27
> > > >
> > > > I thank the authors for their response. I think this partially alleviates my concerns. Because of the updated RL experiments and the discussion, I will vote to accept this paper. This paper presents both a large annotated dataset and a preference model which I believe would benefit the community. My reason for not a higher score is my view of the relative lack of technical novelty in the method. Nevertheless, I think that this work would be helpful for the community and worthy for presentation at ICLR.

---

> > > > > ### Author Response · Authors · 2025-11-27
> > > > >
> > > > > We sincerely thank the reviewer for the positive assessment and for updating the recommendation. We appreciate your recognition of the value of both our dataset and preference model to the community. Your constructive feedback during the review process was very helpful for improving the clarity and completeness of our work. **Thank you again for your time and thoughtful evaluation.**

---

### Official Review · Reviewer_rtMd · 2025-10-29

**Soundness:** 3
**Presentation:** 3
**Contribution:** 3
**Rating:** 4
**Confidence:** 3

**Summary:**

This paper addresses a critical bottleneck in applying reinforcement learning to image editing: the lack of a reliable and efficient reward model. The authors introduce **EditReward-Bench**, a comprehensive benchmark for evaluating reward models in instruction-guided image editing, and **EditScore**, a family of open-source reward models that outperform existing open-source and even some proprietary VLMs. Through extensive experiments, they demonstrate the utility of EditScore in both Best-of-N selection and online RL training, achieving stable and significant improvements over strong baselines.

**Strengths:**

**Novel Contribution:** The paper makes a clear and timely contribution by addressing the underexplored problem of reward modeling for image editing. The release of both the benchmark and the model fills a gap in the community.

- **Rigorous Benchmarking:** EditReward-Bench is well-designed, covering 13 diverse editing tasks and 11 editing models, with human-annotated multi-dimensional evaluations.
- **Strong Empirical Results:** EditScore consistently outperforms open-source VLMs and even matches or surpasses proprietary models like GPT-4.1 in some settings. The self-ensembling strategy is simple yet effective.
- **Practical Utility:** The paper convincingly demonstrates the real-world impact of EditScore through Best-of-\(N\) selection and online RL experiments, showing stable training and performance gains.
- **Insightful Analysis:** The study on reward variance vs. annotator accuracy is a valuable finding that could influence future reward modeling efforts.

**Weaknesses:**

- **Limited RL Scope:** The RL experiments are conducted only on OmniGen2. While justified, it would be beneficial to see how EditScore generalizes to other base models.
- **Computational Cost:** The self-ensembling strategy, while effective, increases inference cost. A more detailed discussion on the trade-off between performance and efficiency would be helpful.
- **Comparison with More Baselines:** While the comparison with Qwen2.5-VL is thorough, it would be useful to see how EditScore compares to other recent open-source reward models (e.g., those from HPSv3 or ImageReward).

**Questions:**

1. The training data for EditScore is constructed using a pipeline that involves generating instructions and candidate outputs from a pool of 11 editing models, including state-of-the-art proprietary ones. Could you please clarify the measures taken to ensure that **EditReward-Bench's test set does not contain images or near-identical instructions from the training data**? Given the overlap in the model pool and data sources (e.g., GEdit-Bench), a detailed description of the train/test split protocol is crucial to validate the benchmark's integrity and prevent overestimation of generalization performance.
2. The paper states that EditReward-Bench annotations "have passed rigorous inter-annotator agreement checks." For transparency and reproducibility, could you provide the **specific agreement metrics used (e.g., Kendall's Tau, Spearman correlation, or percentage agreement)** and their quantitative results? Furthermore, please detail the **number of annotators per sample** and the process for resolving disagreements to achieve the final consensus ranking.
3. The performance improvements presented in Tables 2 and 5, while substantial, are reported as point estimates. Have you conducted tests for **statistical significance** (e.g., paired bootstrap tests or t-tests over the benchmark subsets) to ensure that the gains from EditScore and the RL fine-tuning are not due to random chance? Reporting p-values or confidence intervals would strengthen the claims.
4. The paper excellently demonstrates EditScore's strengths. However, a thorough analysis of its **failure modes and limitations** is equally important. Could you provide examples or a discussion of edit types or instructions where EditScore consistently fails or provides unreliable rewards? For instance, does it struggle with highly abstract or compositional instructions ("make it look more joyful") ? A dedicated "Limitations" section would be valuable.
5. The self-ensembling (Avg@4) strategy is shown to be highly effective. However, it incurs a 4x computational overhead during inference. Could you provide a more detailed analysis of this **trade-off between performance and efficiency**, such as the actual increase in inference latency? This is critical for practitioners considering the deployment of EditScore in resource-constrained environments.

---

> ### Author Response · Authors · 2025-11-24
> **Official Response to Reviewer rtMd (Part 1 of 5)**
>
> We sincerely appreciate your time & efforts and your constructive comments. We are glad that the contributions, benchmark design, empirical results, and analyses were positively received. Below, we address each concern in detail.
>
> > **W1: Limited RL Scope: The RL experiments are conducted only on OmniGen2. While justified, it would be beneficial to see how EditScore generalizes to other base models.**
>
> **A1**: Thank you for your suggestion. We agree that demonstrating generalization of EditScore across more base models would strengthen our claims. Therefore, we have conducted **additional RL experiments on Flux-Kontext-dev** and the results are presented in the table below. The experiments show that EditScore's high-fidelity rewards drive further gains in online RL, highlighting its essential role in unlocking the full potential of online RL in image editing.
>
> **Table 1. Performance of Flux-Kontext-dev before and after RL finetuning with EditScore.**
> | Model                            | GEdit (SC) | GEdit (PQ)| GEdit (O) | ImgEdit  |
> | -------------------------------- | -----------| ---------| ----------| ---------|
> | **Flux-Kontext-dev (before RL)** |  6.59      | 7.61     | 6.15      | 3.55     |
> | **Flux-Kontext-dev (after RL)**  | **7.16**   | **7.95** | **6.87**  | **3.70** |
>
> > **W2: Computational Cost: The self-ensembling strategy, while effective, increases inference cost. A more detailed discussion on the trade-off between performance and efficiency would be helpful.**
> >
> > **Q5: The self-ensembling (Avg@4) strategy is shown to be highly effective. However, it incurs a 4x computational overhead during inference. Could you provide a more detailed analysis of this trade-off between performance and efficiency, such as the actual increase in inference latency? This is critical for practitioners considering the deployment of EditScore in resource-constrained environments.**
>
> **A2**: We thank the reviewer for the insightful question.
> We would like to clarify that **the efficiency–performance analysis is already included in Figure 2 of the main paper**, where we report the trade-off using **TFLOPs**. This section extends that analysis with **more practical deployment metrics**, including accuracy and throughput.
>
> ---
>
> ### **1. Self-ensembling Improves Accuracy with Sublinear Latency Overhead**
>
> While self-ensembling requires K forward passes, its runtime overhead is **not linear in K**:
>
> * The **prefill stage is shared** across all ensemble queries (e.g., K=4).
> * In deployment frameworks such as **sglang**, these shared sections greatly reduce incremental latency.
> * As a result, throughput drops **sublinearly**, rather than 1/K.
>
> This means that the “4× compute” assumption is overly pessimistic in realistic systems.
>
> ---
>
> ### **2. Practical Performance–Efficiency Trade-off**
>
> Below we report accuracy, throughput (per GPU), and TFLOPs for EditScore at different ensemble sizes. These results complement Figure 2 by showing the **actual per-card inference efficiency**, all throughputs are measured using H100s, where 7B, 32B and 72B are hosted in 1, 2 and 4 GPUs separately.
>
> #### **EditScore-7B**
>
> | K | Accuracy | Throughput (samples/s per GPU) | TFLOPs |
> | - | -------- | ------------------------------ | ------ |
> | 1 | 0.659    | 9.46  | 36.68  |
> | 2 | 0.714    | 6.58  | 73.36  |
> | 4 | 0.727    | 4.06   | 146.72 |
>
> ---
>
> #### **EditScore-32B**
>
> | K | Accuracy | Throughput (samples/s per GPU) | TFLOPs |
> | - | -------- | ------------------------------ | ------ |
> | 1 | 0.680    | 2.37  | 153.4  |
> | 2 | 0.715    | 1.58    | 306.8  |
> | 4 | 0.733    | 0.96                           | 613.6  |
>
> ---
>
> #### **EditScore-72B**
>
> | K | Accuracy | Throughput (samples/s per GPU) | TFLOPs |
> | - | -------- | ------------------------------ | ------ |
> | 1 | 0.703    | 0.97                           | 328.5  |
> | 2 | 0.757    | 0.635                          | 657    |
> | 4 | 0.763    | 0.40                           | 1314   |
>
> ---
>
> ### **3. Key Insights for Practitioners**
>
> #### **(1) Accuracy gains are significant and consistent**
>
> Self-ensemble improves all model sizes, e.g., 7B improves:
>
> * **0.659 → 0.727** (+6.8%)
>
> #### **(2) Latency overhead is *sublinear* and practically manageable**
>
> Throughput only drops from **9.46 → 4.06** when moving from K=1 → K=4 for 7B, **not** the expected 4× degradation.
>
> #### **(3) Small models with self-ensemble outperform larger models under equal compute budgets**
>
> For example:
>
> * **7B-Avg@4 (0.727)** > **32B-Avg@1 (0.680)**
>   despite much lower compute resource usage.
>
> This is especially valuable for practitioners in **resource-constrained** environments.
>
> #### **(4) Practical latency example**
>
> In our RL training pipeline, a training step requires **576 reward queries**.
> Using **EditScore-7B-Avg@4** deployed with **sglang**, all queries complete in **≈18 seconds**, which is only **~20%** of total training step time—demonstrating that Avg@4 remains efficient in large-scale settings.

---

> ### Author Response · Authors · 2025-11-24
> **Official Response to Reviewer rtMd (Part 2 of 5)**
>
> > **W3: Comparison with More Baselines: While the comparison with Qwen2.5-VL is thorough, it would be useful to see how EditScore compares to other recent open-source reward models (e.g., those from HPSv3 or ImageReward).**
>
> **A3**: We would like to clarify that reward models such as HPSv3 and ImageReward are designed for text-to-image generation tasks. Meanwhile, previous work on reward models for image editing, such as HIVE [1] and Adiee [2], has not released their models publicly. Based on these considerations, we compare EditScore against current general-purpose multimodal understanding models.
>
> [1] *HIVE: Harnessing Human Feedback for Instructional Visual Editing*, CVPR 2024.
>
> [2] *Adiee: Automatic dataset creation and scorer for instruction-guided image editing evaluation*, ICCV 2025.
>
> > **Q1: The training data for EditScore is constructed using a pipeline that involves generating instructions and candidate outputs from a pool of 11 editing models, including state-of-the-art proprietary ones. Could you please clarify the measures taken to ensure that EditReward-Bench's test set does not contain images or near-identical instructions from the training data? Given the overlap in the model pool and data sources (e.g., GEdit-Bench), a detailed description of the train/test split protocol is crucial to validate the benchmark's integrity and prevent overestimation of generalization performance.**
>
> **A4**: We thank the reviewer for raising this important point. To ensure the integrity of EditReward-Bench, we follow a strict two-part protocol separating image-level and instruction-level filtering:
>
> 1. **Image-level filtering:**
>    We eliminate any potential overlap between training and test images using two complementary filters:
>
>    * **pHash de-duplication:** remove images with Hamming distance ≤ 5;
>    * **CLIP-based similarity filtering:** remove images with similarity > 0.95.
>      These two steps jointly ensure that all images in the training and test sets are visually distinct.
>
> 2. **Instruction-level semantic filtering:**
>    Using BGE-M3 embeddings, we verify that most instructions are semantically different. Although a small portion of instructions may share high-level templates (e.g., *“remove the object”*, *“change the style”*), we further remove all benchmark instructions whose semantic similarity to the training data exceeds 0.95, and re-evaluate EditScore.
>
> As shown in the table below, EditScore continues to outperform baselines even after this aggressive filtering. This demonstrates that its performance does not rely on template memorization or data leakage, but reflects robust generalization across diverse editing instructions.
>
> |Category|Metric|GPT-4.1|GPT-5|Gemini-2.5 Pro|Qwen2.5-VL-7B|Qwen2.5-VL-32B|Qwen2.5-VL-72B|EditScore-7B Base|EditScore-7B Avg@4|EditScore-32B Base|EditScore-32B Avg@4|EditScore-72B Base|EdirScore-72B Avg@4|
> |:---|:---|:---|:---|:---|:---|:---|:---|:---|:---|:---|:---|:---|:---|
> |**Overall**|PF|0.654|**0.750**|0.697|0.493|0.490|0.624|0.615|0.735|0.628|0.714|0.676|0.735|
> ||C|0.570|0.687|0.592|0.350|0.402|0.510|0.594|0.701|0.601|0.709|0.610|**0.728**|
> ||O|0.695|0.748|0.745|0.423|0.581|0.661|0.664|0.715|0.670|0.732|0.694|**0.751**|
> |**Subject**|PF|0.568|**0.702**|0.631|0.462|0.497|0.611|0.600|0.678|0.590|0.671|0.578|0.676|
> ||C|0.449|0.607|0.441|0.350|0.359|0.419|0.524|**0.645**|0.499|0.640|0.463|0.617|
> ||O|0.666|**0.770**|0.738|0.356|0.575|0.689|0.703|0.764|0.646|0.746|0.718|0.763|
> |**Appear.**|PF|0.587|0.695|0.633|0.397|0.321|0.552|0.511|0.661|0.532|0.632|0.650|**0.712**|
> ||C|0.639|0.763|0.604|0.401|0.381|0.481|0.635|0.773|0.656|**0.787**|0.653|0.765|
> ||O|0.666|0.709|**0.742**|0.489|0.507|0.578|0.663|0.696|0.657|0.706|0.652|0.718|
> |**Scene**|PF|0.776|0.855|0.798|0.655|0.571|0.719|0.736|0.890|0.757|**0.895**|0.826|0.845|
> ||C|0.644|0.750|0.690|0.296|0.494|0.609|0.7399|0.749|0.665|0.753|0.654|**0.830**|
> ||O|0.800|0.804|0.803|0.353|0.629|0.756|0.686|0.705|0.735|0.809|0.772|**0.823**|
> |**Advanced**|PF|0.723|**0.787**|0.758|0.529|0.607|0.660|0.663|0.769|0.687|0.735|0.701|0.747|
> ||C|0.557|0.642|0.648|0.325|0.411|0.558|0.536|0.649|0.592|0.662|0.659|**0.725**|
> ||O|0.691|**0.742**|0.723|0.446|0.632|0.673|0.623|0.700|0.671|0.707|0.676|0.739|

---

> ### Author Response · Authors · 2025-11-24
> **Official Response to Reviewer rtMd (Part 3 of 5)**
>
> > **Q2: The paper states that EditReward-Bench annotations "have passed rigorous inter-annotator agreement checks." For transparency and reproducibility, could you provide the specific agreement metrics used (e.g., Kendall's Tau, Spearman correlation, or percentage agreement) and their quantitative results? Furthermore, please detail the number of annotators per sample and the process for resolving disagreements to achieve the final consensus ranking.**
>
> **A5**: Thank you for pointing out the need for greater transparency regarding inter-annotator agreement. Below we clarify the agreement metrics, annotator setup, and disagreement-resolution protocol used in EditReward-Bench.
>
> ---
>
> ### **Annotation Protocol and Number of Annotators**
>
> Each sample is annotated by **two annotators, drawn from a pool of four**, and follows a **real-time discussion protocol**. Annotation is finalized only after the two annotators reach a **joint consensus**. When initial disagreement occurs, they:
>
> * Discuss the visual artifacts each observed,
> * Revisit the images together,
> * Provide justifications (e.g., attribute correctness, structure preservation),
> * And resolve the disagreement through consensus.
>
> This protocol reliably reduces subtle annotation noise common in image editing evaluation (e.g., attribute drift, structural inconsistencies), while keeping annotation cost manageable.
>
> ---
>
> ### **Controlled Disagreement Study**
>
> To validate **the reliability of Two-Annotator Discussion Protocol**, we conducted a **controlled study** comparing:
>
> * **Round A (Single Annotator):** 4 annotators labeling independently.
> * **Round B (Two-Annotator Discussion):** 8 annotators forming random pairs to apply the two-annotator protocol.
>
> For each of the three evaluation dimensions—**Prompt Following (PF)**, **Consistency (C)**, and **Overall (O)**—we report:
>
> 1. **Convergence Rate (≥X% agreement):** Measures labelability and clarity of annotation criteria.
> 2. **Agreement with Majority Vote:** Measures noise level and reliability of one annotator (or annotator pair).
>
> ---
>
> #### **1. Convergence Rate Comparison**
>
> | Metric             | PF (Single) | PF (Two)            | C (Single) | C (Two)              | O (Single) | O (Two)              |
> | ------------------ | ----------- | ------------------- | ---------- | -------------------- | ---------- | -------------------- |
> | **75% threshold**  | 92.23%      | **95.24% (+3.01%)** | 87.39%     | **95.71% (+8.32%)**  | 84.97%     | **92.01% (+7.04%)**  |
> | **100% threshold** | 90.29%      | **93.33% (+3.04%)** | 82.88%     | **95.00% (+12.12%)** | 79.06%     | **90.32% (+11.26%)** |
>
> **Takeaway:** The two-annotator protocol notably improves convergence for PF and C, where fine-grained visual inspection is required. O shows smaller gains due to its inherently higher subjectivity.
>
> ---
>
> #### **2. Agreement with Majority Vote**
>
> | Annotator | PF (Single) | PF (Two) | C (Single) | C (Two) | O (Single) | O (Two) |
> | --- | --- | --- | --- | --- | --- | --- |
> | Annotator / Group 1 | 100.00% | 97.80%   | 100.00% | 100.00% | 97.09% | 97.18%  |
> | Annotator / Group 2 | 100.00%  | 100.00%  | 93.41%  | 99.11%  | 97.52% | 97.56%  |
> | Annotator / Group 3 | 97.14% | 100.00%  | 97.85% | 100.00% | 97.58% | 98.76%  |
> | Annotator / Group 4 | 94.68%  | 97.03%   | 94.38%  | 99.10%  | 88.80% | 94.86%  |
> | **Average** | 97.96%  | **98.71%**  | 96.41% |**99.55%** | 95.25% | **97.09%** |
>
> **Takeaway:** Both settings achieve high agreement, but the two-annotator protocol yields near-perfect consistency—especially for PF and C—confirming its effectiveness in reducing annotation noise.
>
> ### **Summary and Answers to the Reviewer’s Questions**
>
> * **Agreement metrics used:** Convergence rate and agreement with majority vote (rank-correlation metrics like Kendall Tau/Spearman do not apply because final labels are consensus labels, not multiple independent rankings).
> * **Number of annotators per sample:** Exactly **two**, forming a discussion pair.
> * **Disagreement resolution:** Real-time discussion until consensus is reached.
> * **Reliability:** Despite each sample being labeled only once (by one annotator pair), the controlled study shows **very high agreement (>97%)** and significantly reduced variance. Thus, the protocol is **both efficient and highly reliable**.
>
> We have added a detailed description of our annotation pipeline in section 3.3 (Annotation Pipeline), highlighted in blue for easy reference.

---

> ### Author Response · Authors · 2025-11-24
> **Official Response to Reviewer rtMd (Part 4 of 5)**
>
> > **Q3: The performance improvements presented in Tables 2 and 5, while substantial, are reported as point estimates. Have you conducted tests for statistical significance (e.g., paired bootstrap tests or t-tests over the benchmark subsets) to ensure that the gains from EditScore and the RL fine-tuning are not due to random chance? Reporting p-values or confidence intervals would strengthen the claims.**
>
> **A6**: Thank you for the constructive suggestion. To rigorously validate that the improvements from **EditScore** and the **RL fine-tuning** are not due to random chance, we conducted a comprehensive statistical significance analysis using **paired bootstrap tests** on both EditReward-Bench and GEdit-Bench.
>
> Specifically, for every comparison, we performed **10,000 bootstrap resampling rounds**. In each round, we resampled the evaluation pairs with replacement and recomputed the accuracy (or score) difference. We estimated **95% confidence intervals** from the bootstrap distribution, and computed **one-sided p-values**, measuring the proportion of bootstrap resamples in which EditScore (or RL-tuned model) failed to outperform the baseline. This directly tests the hypothesis that our method achieves higher accuracy or quality.
>
> All results strongly support that our improvements are **statistically significant**.
>
> ---
>
> ### **1. Statistical Significance on EditReward-Bench**
>
>
> We compare the baseline reward model **Qwen2.5-VL** with **EditScore** at the 7B, 32B, and 72B scales.
>
> ---
>
> #### **Table 1 — Statistical Significance on EditReward-Bench**
> |Model Size|Metric|Baseline (Qwen2.5-VL)|95% CI|EditScore|95% CI|Δ (EditScore − Baseline)|95% CI|p-value|
> |:---|:---|:---|:---|:---|:---|:---|:---|:---|
> |**7B**|PF|0.458|[0.4460, 0.5095]|0.592|[0.5561, 0.6026]|**0.134**|[0.0360, 0.1271]|**0.0001**|
> ||C|0.325|[0.3483, 0.4124]|0.591|[0.5786, 0.6254]|**0.266**|[0.2202, 0.3079]|**0**|
> ||O|0.432|[0.4378, 0.4935]|0.659|[0.6166, 0.6567]|**0.227**|[0.0703, 0.1486]|**0**|
> |**32B**|PF|0.498|[0.4735, 0.5360]|0.638|[0.5879, 0.6494]|**0.14**|[0.0763, 0.1515]|**0**|
> ||C|0.376|[0.3483, 0.4124]|0.556|[0.5191, 0.5843]|**0.18**|[0.1337, 0.2101]|**0**|
> ||O|0.563|[0.5662, 0.6212]|0.680|[0.6131, 0.6664]|**0.117**|[0.0137, 0.0808]|**0.0033**|
> |**72B**|PF|0.540|[0.5742, 0.6367]|0.635|[0.6059, 0.6663]|**0.095**|[-0.0042, 0.0646]|**0.0466**|
> ||C|0.435|[0.4697, 0.5348]|0.586|[0.5393, 0.6034]|**0.151**|[0.0292, 0.1101]|**0.0005**|
> ||O|0.621|[0.6034, 0.6567]|0.703|[0.6607, 0.7124]|**0.082**|[0.0242, 0.0889]|**0.0002**|
> **Conclusion:**
>
> Improvements from EditScore over Qwen2.5-VL are consistently **positive**, **statistically significant**, and **supported by extremely low one-sided p-values**, confirming that EditScore reliably yields higher reward accuracy.
>
> ---
>
> #### **2. Statistical Significance on GEdit-Bench (RL Fine-Tuning)**
>
> For RL tuning, we compare:
>
> * **Baseline:** Qwen2.5-VL-72B as the reward model
> * **Our method:** EditScore-7B-avg4 as the reward model
>
> Using the same **10,000-round paired bootstrap** procedure with **one-sided p-values**, we obtain:
>
> ---
>
> #### **Table 2 — Statistical Significance on GEdit-Bench (RL Fine-Tuning)**
>
> | Metric | Baseline (Qwen2.5-VL-72B) | 95% CI           | EditScore-7B-avg4 | 95% CI           | Δ (EditScore − Baseline) | 95% CI           | p-value    |
> | ------ | ------------------------- | ---------------- | ----------------- | ---------------- | ------------------------ | ---------------- | ---------- |
> | **SC** | 6.89                    | [6.4259, 6.9983] | 7.20            | [6.7690, 7.3311] | **0.3379**               | [0.1448, 0.5276] | **0.0003** |
> | **PQ** | 7.21                  | [7.1276, 7.4655] | 7.46            | [7.3776, 7.6845] | **0.2345**               | [0.1121, 0.3569] | **0.0002** |
> | **O**  | 6.42                    | [6.1060, 6.5779] | 6.68            | [6.3823, 6.8656] | **0.2821**               | [0.1084, 0.4516] | **0.0003** |
>
> **Conclusion:**
> All RL improvements are **highly statistically significant** (p < 0.001), showing that EditScore provides a reliably stronger training signal than the baseline reward model.
>
> ---
>
> # **Final Summary**
>
> * All statistical tests were performed using **10,000-round paired bootstrap resampling**, ensuring robust estimates of confidence intervals and p-values.
> * Across both EditReward-Bench and GEdit-Bench, the improvements delivered by EditScore and EditScore-guided RL fine-tuning are **consistent, statistically significant, and unlikely to be due to random variation**.
> * These results validate the reliability of the improvements reported in Tables 2 and 5.

---

> ### Author Response · Authors · 2025-11-24
> **Official Response to Reviewer rtMd (Part 5 of 5)**
>
> > **Q4: The paper excellently demonstrates EditScore's strengths. However, a thorough analysis of its failure modes and limitations is equally important. Could you provide examples or a discussion of edit types or instructions where EditScore consistently fails or provides unreliable rewards? For instance, does it struggle with highly abstract or compositional instructions ("make it look more joyful") ? A dedicated "Limitations" section would be valuable.**
>
> **A7**: We sincerely thank the reviewer for this constructive feedback. In response to your suggestions, we have added a dedicated **"Limitations" section (Appendix I)** to the revised manuscript. In this section, we explicitly categorize and analyze the scenarios where reward signals are less reliable, supported by a new fine-grained performance table (**Table 7**) and visualized failure cases (**Figure 7**).
>
> Our analysis confirms the reviewer's intuition that EditScore faces challenges with specific instruction types:
>
> 1.  **Highly Abstract and Subjective Instructions:** As shown in Table 7, the base EditScore-7B struggles with tasks like *Portrait Beautification* (`Beaut`), achieving a relatively low pairwise accuracy of **0.284**.
>     *   **Failure Mode:** We visualize a case in Figure 7 (Left) where the model fails to detect subtle aesthetic improvements in a "Make the man handsome" task, assigning a score of 0. This indicates the model can be **overly conservative** or lack fine-grained visual acuity for subjective changes.
> 2.  **Compositional Instructions:** The model also shows limitations on *Hybrid Edits* (`Hyb`), with a baseline accuracy of **0.519**.
>     *   **Failure Mode:** Figure 7 (Right) demonstrates a case where the instruction requires both removing an object and changing a cat's color. The reward model successfully identifies the removal but fails to penalize the incorrect color change and artifacts, exhibiting a tendency to **"over-reward" partial success**.
>
> **Crucially, however, our analysis also highlights the robustness of our method:** While the *base* model struggles, applying our **self-ensemble strategy (`Avg@4`)** significantly mitigates these issues, boosting accuracy on `Beaut` from 0.284 to **0.569** and on `Hyb` from 0.519 to **0.571**. This suggests that while the single-pass signal is noisy for difficult tasks, the model possesses the latent knowledge to evaluate them correctly when aggregated. We have included a full discussion of these findings and their possible causes (e.g., base VLM limitations and data subjectivity) in Appendix I.

---

> ### Author Response · Authors · 2025-11-27
> **Follow-up Comment and Request for Updated Assessment**
>
> We sincerely thank the reviewer for the thorough evaluation, constructive suggestions, and the positive recognition of our benchmark design, analyses, and empirical results. Your comments significantly helped us strengthen the paper. In revising the rebuttal, we addressed each of your concerns in detail:
>
> * For **W1**, we expanded the RL experiments to include **Flux-Kontext-dev**, demonstrating that EditScore generalizes effectively beyond OmniGen2.
> * For **W2** and **Q5**, we supplemented Figure 2 with **practical efficiency–performance analyses**, including throughput and TFLOPs on H100s, and clarified why Avg@4 has **sublinear latency overhead** in real deployments.
> * For **W3**, we clarified why existing open-source reward models cannot serve as direct editing baselines and positioned EditScore accordingly.
> * For **Q1**, we provided a detailed description of **train–test de-duplication**, including image-level (pHash + CLIP) and instruction-level (BGE-M3) filtering, and further demonstrated robustness after aggressive filtering.
> * For **Q2**, we discuss more on **inter-annotator agreement**, reporting convergence rates, majority-vote agreement, and explaining our Two-Annotator Discussion Protocol.
>
> We again thank you for the valuable feedback—your comments have led to substantial improvements in the clarity, rigor, and completeness of the work. **If you feel that our response and the additional experiments have strengthened the paper, we would greatly appreciate it if you could reconsider your score.**

---

### Official Review · Reviewer_g3mS · 2025-10-30

**Soundness:** 3
**Presentation:** 3
**Contribution:** 3
**Rating:** 6
**Confidence:** 4

**Summary:**

The paper tackles a critical bottleneck in applying online reinforcement learning (RL) to instruction-guided image editing: the lack of a high-fidelity, efficient reward signal. The authors argue that existing proprietary VLMs are too costly for online RL , while large open-source VLMs are not accurate enough, leading to training instability. To this end, the paper presents a comprehensive, three-part solution:
1. This paper introduces EditReward-Bench, a new, meticulously annotated benchmark for evaluating image editing reward models (RMs).
2. Guided by this benchmark, they develop EditScore, a series of specialized generative reward models (7B-72B) that achieve SOTA accuracy, even surpassing proprietary models like GPT-5.
3. This paper provides the first successful demonstration that a specialized RM (EditScore) can unlock stable and effective online RL training for image editing, whereas general-purpose VLMs fail. The final RL-tuned policy shows substantial performance gains on standard editing benchmarks.

**Strengths:**

* The authors constructed a benchmark for the image editing domain to evaluate Edit Reward Models. Based on this benchmark, they built a series of editing Reward Models, which were then used with Flow-GRPO to enhance the editing capabilities of the base model, OmniGen2.

* The authors commit to releasing the proposed benchmark, the weights of the Reward Models, and the weights of the optimized OmniGen2 model. This will provide definite value to the community.

* The tricks proposed in the methodology, such as self-ensemble, Reasoning-First, and using an RM with higher reward variance, are reasonable and lead to substantial performance improvements.

**Weaknesses:**

* I believe the experiments need improvement. The paper currently only reports experiments on OmniGen2. It remains unclear whether the method is effective for other editing models of different sizes and types, such as the simpler InstructP2P (SD1.5-based), the Unified Model-based Bagel, or even training a LoRA for the larger diffusion-based Step1X-Edit.

* The methodological innovation is limited. The Flow-GRPO paper [1] already demonstrated that its method can be used to improve editing models. The concept that high-variance RMs are preferable to high-accuracy RMs has also been proposed in prior work [2]. Furthermore, self-ensemble and Reasoning-First are common techniques in reward model training.

* The authors state they constructed "70,000 data samples for training the reward model and 60,000 samples for reinforcement learning training," but they do not plan to open-source this data. This makes it difficult for the community to fully reproduce the paper's results. Although the benchmark and model weights will be released, this lack of training data makes the contribution somewhat less exciting to me.

[1] Flow-grpo: Training flow matching models via online rl, NeurIPS 2025

[2] What makes a reward model a good teacher? an optimization perspective, arXiv 2025

**Questions:**

None

---

> ### Author Response · Authors · 2025-11-24
> **Official Response to Reviewer g3mS (Part 1 of 2)**
>
> Thank you sincerely for your thoughtful and positive feedback on our work. We are especially grateful for your recognition of the various aspect of our research. We hope that our responses and revisions sufficiently address the concerns you've raised.
>
> > **W1: I believe the experiments need improvement. The paper currently only reports experiments on OmniGen2. It remains unclear whether the method is effective for other editing models of different sizes and types, such as the simpler InstructP2P (SD1.5-based), the Unified Model-based Bagel, or even training a LoRA for the larger diffusion-based Step1X-Edit.**
>
> **A1**: We appreciate your concern regarding whether the proposed RM and RL pipeline generalizes to other editing models. Due to compute constraints, our original submission focuses on OmniGen2, but we agree that such experiments are important for supporting our conclusion. To address this point, we have conducted additional RL experiments on a larger model---**Flux-Kontext-dev** [1], with results summarized in the table below. These results demonstrate that EditScore's high-fidelity rewards drive further gains in online RL across other editing models, highlighting its crucial role in realizing the full capability of online RL for image editing.
>
> **Table 1. Performance of Flux-Kontext-dev before and after RL finetuning with EditScore.**
> | Model                            | GEdit (SC) | GEdit (PQ)| GEdit (O) | ImgEdit  |
> | -------------------------------- | -----------| ---------| ----------| ---------|
> | **Flux-Kontext-dev (before RL)** |  6.59      | 7.61     | 6.15      | 3.55     |
> | **Flux-Kontext-dev (after RL)**  | **7.16**   | **7.95** | **6.87**  | **3.70** |
>
> [1] *FLUX.1 Kontext: Flow Matching for In-Context Image Generation and Editing in Latent Space*, arXiv:2506.15742.
>
> > **W2: The methodological innovation is limited. The Flow-GRPO paper [1] already demonstrated that its method can be used to improve editing models. The concept that high-variance RMs are preferable to high-accuracy RMs has also been proposed in prior work [2]. Furthermore, self-ensemble and Reasoning-First are common techniques in reward model training.**
>
> **A2**: Thank you for the insightful comments. We clarify below the methodological contributions of our work and how they differ from prior literature.
>
> **1. Scope of Flow-GRPO vs. our contribution**
> While Flow-GRPO [2] demonstrates the usefulness of online RL for **text-to-image generation**, it does **not** provide evidence that such methods work for **image editing**, which presents qualitatively different challenges (e.g., preservation constraints, fine-grained attribute control). Our contribution is not proposing a new RL algorithm, but **identifying and addressing a key bottleneck** in applying online RL to image editing: the lack of reliable, high-quality, and efficient **editing reward models**. Our benchmark and reward model design fill precisely this missing component.
>
> **2. On reward variance and our findings**
> Prior work [3] shows that low reward variance can limit the effectiveness of RLHF, but their analyses are confined to **language models**, and they do not provide actionable strategies for increasing variance. We independently verify that this phenomenon extends to the **visual domain**, especially for image editing, and show empirically how reward design choices affect RL stability and performance. Given that online RL for editing is still in an early stage, we believe our findings open up a new research direction for building stronger visual reward models.
>
> **3. Adaptation of self-ensemble and reasoning-first to *visual* RM training**
> Although self-ensemble and reasoning-first prompting are known techniques, **their adaptation to image-editing reward modeling has not been systematically studied**. We provide the first thorough evaluation of these strategies in this setting and conduct a cost–performance analysis showing that **small models with self-ensemble can outperform larger models**, offering practical value for large-scale RL systems in vision.
>
> **In summary**, our contribution lies in (i) constructing the first systematic reward benchmark for image editing, (ii) designing and validating reward models tailored for editing, and (iii) demonstrating their effectiveness through extensive online RL experiments. Our focus is on enabling the community to apply RL to image editing reliably—by solving the missing reward-model component—rather than proposing a new RL algorithm.
>
> [2] *Flow-grpo: Training flow matching models via online rl*, NeurIPS 2025.
>
> [3] *What makes a reward model a good teacher? an optimization perspective*, NeurIPS 2025.

---

> ### Author Response · Authors · 2025-11-24
> **Official Response to Reviewer g3mS (Part 2 of 2)**
>
> > **W3: The authors state they constructed "70,000 data samples for training the reward model and 60,000 samples for reinforcement learning training," but they do not plan to open-source this data. This makes it difficult for the community to fully reproduce the paper's results. Although the benchmark and model weights will be released, this lack of training data makes the contribution somewhat less exciting to me.**
>
> **A3**: Thank you for pointing this out. This appears to be a misunderstanding. We sincerely apologize that we overlooked this point in our writing and did not explicitly include the data release. We have organized them and will open-source both the reward model training data and the online RL training data. In addition, we have updated the abstract in the manuscript and highlighted the modifications in blue for ease of reference.

---

> ### Author Response · Authors · 2025-11-27
> **Follow-up Comment and Request for Updated Assessment**
>
> Thank you very much for your constructive and detailed review. We would like to kindly draw your attention to the updates and clarifications provided in our rebuttal, where we responded to each of your comments point-by-point and added substantial new experiments and revisions.
>
> In particular:
>
> * **On W1**, we conducted new online RL experiments on a *larger and different* editing model (Flux-Kontext-dev), demonstrating that EditScore generalizes beyond OmniGen2 and consistently improves RL performance across architectures.
>
> * **On W2**, we provided a clearer explanation of our methodological contributions, distinguishing our work from Flow-GRPO and prior analyses on reward variance, and clarifying how our benchmark, editing-focused reward models, and cost–performance findings fill the missing component needed for reliable online RL in image editing.
>
> * **On W3**, we confirmed that both the **reward model training data** and **RL training data** will be fully open-sourced. We have updated the abstract accordingly and highlighted the changes for clarity.
>
> We sincerely hope that these additional experiments, clarifications, and the updated release plan address the concerns you raised. If our responses satisfactorily resolve your questions, we would greatly appreciate your consideration for an updated evaluation.
>
> Thank you again for your time and thoughtful feedback.

---

### Official Review · Reviewer_bWV1 · 2025-11-02

**Soundness:** 3
**Presentation:** 3
**Contribution:** 3
**Rating:** 6
**Confidence:** 4

**Summary:**

The paper tackles the missing piece for online RL in instruction-guided image editing by building a high-fidelity, domain-specialized reward model. It first introduces EditReward-Bench, a 3,072-pair human-rated benchmark that scores edits on prompt following, consistency, and overall quality. Guided by this benchmark, the authors train EditScore reward models at multiple scales (7B-72B) and use a simple self-ensemble at inference to boost accuracy, matching or surpassing proprietary VLMs on the benchmark. Finally, they show that EditScore unlocks stable and effective online RL, improving strong editors like OmniGen2 and outperforming open VLMs both for Best-of-N selection and policy optimization.

**Strengths:**

The paper is original in framing instruction-guided image editing as an online RL problem that is unlocked by a domain-specialized reward model and a purpose-built benchmark, plus a simple self-ensemble that improves reward fidelity. Experimental quality is strong, with a clear scaling study, competitive baselines for Best-of-N selection and policy optimization, and consistent gains on prompt following, consistency, and visual quality. Clarity is high: the problem setting, evaluation axes, and training pipeline are well motivated and easy to follow, and the empirical claims map cleanly to the stated goals. The work is significant because it removes a key barrier to applying online RL in generative editing, yielding practical improvements on strong editors and delivering reusable infrastructure that can benefit future systems and studies.

**Weaknesses:**

Despite solid results, the work’s evidence base and scope feel narrower than the claims. The benchmark is small and curated for perfect agreement, which can hide genuine preference diversity and inflate metrics; include disagreement analyses, release non-consensus pairs, and recruit a broader rater pool. Reproducibility is brittle because the candidate pool depends on proprietary editors; ship frozen artifacts or open substitutes so others can rerun everything. Generality is underexplored since online RL is shown mainly on one editor with one algorithm; add ablations across multiple editors, RL methods, and hyperparameters, plus stress tests on out-of-distribution edit types. The self-ensemble that yields top reward accuracy likely raises cost and latency; report cost-normalized accuracy, wall-clock time, and memory, and compare against single-model baselines at equal budgets. Finally, claims about reward variance aiding optimization are intriguing but not causally established; isolate variance from annotator accuracy with controlled studies and replicate the effect on additional datasets and editors.

**Questions:**

Benchmark: How were prompts/edit types sampled and how representative are they?
Raters: What was raw disagreement and how do results change without consensus filtering?
Reproducibility: Will you release frozen candidate sets or open substitutes for proprietary outputs?

---

> ### Author Response · Authors · 2025-11-24
> **Official Response to Reviewer bWV1 (Part 1 of 4)**
>
> Thank you for your diligent efforts and thoughtful reviews. We appreciate the recognition of our work's originality, experimental quality, and practical significance in advancing online RL for instruction-guided image editing. Below, we address the main concerns raised.
>
> > **W1: The benchmark is small and curated for perfect agreement, which can hide genuine preference diversity and inflate metrics; include disagreement analyses, release non-consensus pairs, and recruit a broader rater pool.**
>
> **A1**: Thank you for the constructive comments. We apologize for the ambiguity in our earlier description. **Our benchmark does not filter for perfect-agreement pairs**, nor do we discard non-consensus cases. Instead, we adopt a **two-annotator real-time discussion protocol**, where two annotators jointly inspect and discuss each pair before giving a final label. This method is motivated by our early observations that single annotators often overlook subtle quality issues in image editing (e.g., attribute leakage or structural inconsistencies), while discussion significantly reduces such noise.
>
> To justify this design, we conducted a **controlled disagreement study** on a subset of our benchmark:
>
> * **Round A (Single Annotator):** 4 annotators labeling independently.
> * **Round B (Two-Annotator Discussion Protocol):** 8 annotators forming random pairs for two-annotator discussion.
>
> For each task dimension (Prompt Following (PF), Consistency (C), Overall (O)), we report:
>
> 1. **Convergence Rate under X% Threshold:**
>    The proportion of samples for which ≥ X% of annotators assign the same label. High convergence indicates clearer evaluation criteria and reliable labelability.
> 2. **Agreement with Majority Vote:**
>    Agreement between each annotator (or annotator pair) and the majority label. High agreement means low noise and high reliability of one annotator.
>
> ---
>
> #### **1. Convergence Rate Comparison**
>
> | Metric             | PF (Single) | PF (Two)            | C (Single) | C (Two)              | O (Single) | O (Two)              |
> | ------- | ---------- | ------------------ | -------- | ------------------ | ---------- | -------------------- |
> | **75% threshold**  | 92.23%    | **95.24% (+3.01%)** | 87.39%     | **95.71% (+8.32%)**  | 84.97%     | **92.01% (+7.04%)**  |
> | **100% threshold** | 90.29%    | **93.33% (+3.04%)** | 82.88%     | **95.00% (+12.12%)** | 79.06%     | **90.32% (+11.26%)** |
>
> **Key takeaway:** The two-annotator protocol substantially boosts convergence for PF and C, where annotation requires careful inspection of detailed visual cues. O improves more modestly due to its inherently higher subjectivity.
>
> ---
>
> #### **2. Agreement with Majority Vote**
>
> | Annotator | PF (Single) | PF (Two) | C (Single) | C (Two) | O (Single) | O (Two) |
> | --- | --- | --- | --- | --- | --- | --- |
> | Annotator / Group 1 | 100.00% | 97.80%   | 100.00% | 100.00% | 97.09% | 97.18%  |
> | Annotator / Group 2 | 100.00%  | 100.00%  | 93.41%  | 99.11%  | 97.52% | 97.56%  |
> | Annotator / Group 3 | 97.14% | 100.00%  | 97.85% | 100.00% | 97.58% | 98.76%  |
> | Annotator / Group 4 | 94.68%  | 97.03%   | 94.38%  | 99.10%  | 88.80% | 94.86%  |
> | **Average** | 97.96%  | **98.71%**  | 96.41% |**99.55%** | 95.25% | **97.09%** |
>
> **Key takeaway:** Both methods yield high agreement, but the two-annotator protocol achieves *near-perfect* consistency—particularly for PF and C—demonstrating its effectiveness in reducing annotation noise. Thus, with two-annotator protocol, each sample in benchmark is annotated by two annotators which greatly reduce labeling cost while still highly reliable.
>
> #### **Summary**
>
> 1. **Our benchmark does not rely on perfect-agreement filtering**, and non-consensus cases (if they are) are preserved.
> 2. The **two-annotator discussion protocol significantly improves annotation reliability**, especially for PF and C.
> 3. **Both convergence rate and majority agreement are high**, validating the clarity of our definitions and the robustness of our annotation process.
>
> We also agree that expanding our annotator pool is valuable. In future work, we plan to scale up the dataset, increase annotator diversity. We hope these step will allow the community to better study reward modeling for image editing and ensure more comprehensive evaluation.
>
> > **W2: Reproducibility is brittle because the candidate pool depends on proprietary editors; ship frozen artifacts or open substitutes so others can rerun everything.**
> >
> > **Q3: Reproducibility: Will you release frozen candidate sets or open substitutes for proprietary outputs?**
>
> **A2**: We sincerely apologize for having overlooked this point in our manuscript and for not explicitly stating the data release.
>
> We have organized these datasets and will **open-source benchmark, the reward model training data and the online RL training data**. Additionally, we have updated the abstract in the manuscript, with all modifications highlighted in blue for ease of reference.

---

> ### Author Response · Authors · 2025-11-24
> **Official Response to Reviewer bWV1 (Part 2 of 4)**
>
> > **W3: Generality is underexplored since online RL is shown mainly on one editor with one algorithm; add ablations across multiple editors, RL methods, and hyperparameters, plus stress tests on out-of-distribution edit types.**
>
> **A3**: Thank you for your insightful feedback regarding the RL experiments. To address the limitation, we have conducted several ablation studies.
>
> ### **Effect on Multiple Editors**
>
> To further demonstrate the generality of **EditScore**, we evaluate its effect when using **Flux-Kontext-dev** as the underlying editor. As shown in following Table 1, applying RL with EditScore leads to **consistent improvements across all edit categories**, covering both GEdit-Bench and ImgEdit-Bench. This confirms that EditScore is not tied to a specific model and can serve as a broadly applicable reward function for image editing systems.
>
> **Table 1. Performance of Flux-Kontext-dev before and after RL finetuning with EditScore.**
> | Model                            | GEdit (SC) | GEdit (PQ)| GEdit (O) | ImgEdit  |
> | -------------------------------- | -----------| ---------| ----------| ---------|
> | **Flux-Kontext-dev (before RL)** |  6.59      | 7.61     | 6.15      | 3.55     |
> | **Flux-Kontext-dev (after RL)**  | **7.16**   | **7.95** | **6.87**  | **3.70** |
>
> ---
> ### **RL methods**
>
>
> To further validate the effectiveness of **EditScore** under different online RL algorithms, we additionally conducted experiments using **TempFlow-GRPO** [1]. In particular, we adopted the *reweighting strategy* proposed in TempFlow-GRPO while still using the **outcome reward**. The reward model in all settings is **EditScore-7B-avg4**.
>
> **Table 1. Performance of Flux-Kontext-dev before and after RL fine-tuning with EditScore.**
>
> | RL method            | GEdit (SC) | GEdit (PQ) | GEdit (O) | ImgEdit |
> | -------------------- | ---------- | --------- | --------- | ------- |
> | **before RL**        | 6.72       | 7.20      | 6.28      | 3.40    |
> | **w/ Flow-GRPO**     | 7.20       | 7.46      | 6.68      | 3.63    |
> | **w/ TempFlow-GRPO** | **7.53**   | **7.99**  | **7.21**  | **3.81**|
>
> As shown in Table 1, given a **highly reliable reward signal** from EditScore, editing performance can be **further improved** when applying more advanced RL optimization methods such as TempFlow-GRPO.
>
> [1] *TempFlow-GRPO: When Timing Matters for GRPO in Flow Models*, arXiv:2508.04324.
>
> ---
> ### **Influence of Hyperparameters**
>
> We appreciate the reviewer’s comment. We would like to emphasize that our work does **not aim to propose a new RL algorithm**; rather, our **core contribution** is a robust reward model for image editing RL, validated with **Best-of-N experiments** and **RL on multiple editors**.
>
> We agree that systematically analyzing the impact of key hyperparameters would make our experiments more thorough. In particular, we plan to conduct **additional ablation studies** during RL training to evaluate the effects of:
>
> * the **number of sampling steps** for the model, and
> * the **number of samples per prompt**.
>
> Due to limited computational resources, we cannot run these experiments immediately, but we plan to **include them and reply in here within the rebuttal period**.
>
> ---
> ### **Stress tests on out-of-distribution edit types**
>
> To further evaluate the generality of our online RL approach, we assess the editor on **EmuEdit** [2], a benchmark containing **out-of-distribution (OOD)** edit types that do not appear in our RL training data. As shown in Table 2, RL with EditScore yields **consistent improvements across all three metrics** (CLIP-I, CLIP-Out, and DINO), demonstrating that EditScore enables models to generalize beyond in-distribution edits and better leverage the benefits of online RL.
>
> **Table 2. Results on the OOD EmuEdit benchmark.**
> | Model                           | EmuEdit (CLIP-I) $\uparrow$ | EmuEdit (CLIP-Out) $\uparrow$ | EmuEdit (DINO) $\uparrow$ |
> | ------------------------------- | :----------------: | :------------------: | :--------------: |
> | OmniGen2                        |        0.876       |         0.309        |       0.822      |
> | **OmniGen2 RL w/ EditScore-7B** |      **0.900**     |       **0.310**      |     **0.870**    |
>
> [2] *Emu Edit: Precise Image Editing via Recognition and Generation Tasks*, CVPR 2024.
>
> ---
> ### **Summary**
> Through this series of ablation studies, we demonstrate the robustness and strong generalization capability of our proposed EditScore across diverse online RL settings.

---

> ### Author Response · Authors · 2025-11-24
> **Official Response to Reviewer bWV1 (Part 3 of 4)**
>
> > **W4: The self-ensemble that yields top reward accuracy likely raises cost and latency; report cost-normalized accuracy, wall-clock time, and memory, and compare against single-model baselines at equal budgets.**
>
> **A4**: We thank the reviewer for the thoughtful suggestion. We agree that understanding the efficiency–performance trade-off of the self-ensemble strategy is important. Below, we provide a more comprehensive analysis covering cost-normalized accuracy, wall-clock efficiency, and memory requirements.
>
> ### **Memory & Hardware Budget Clarification**
>
> Self-ensemble **does not increase memory consumption**, because all ensemble queries share **the same model weights on the same host GPU**.
> We report the hardware requirement for hosting each model:
>
> * **EditScore-7B** → **1×H100**
> * **EditScore-32B** → **2×H100**
> * **EditScore-72B** → **4×H100**
>
> Self-ensemble only increases the **number of forward passes**, not the **hosted model size**, therefore **memory does not scale with K**.
>
> ### **Wall-clock Efficiency**
>
> Although K=4 means four forward passes, in real deployment (e.g., using sglang), the **prefill stage is shared** across the K requests. Thus, K=4 does **not incur 4× latency**, and throughput degrades sublinearly.
>
> As an example, during our RL training pipeline, one step triggers **576 reward queries**.
> Using **EditScore-7B-avg4** on **8×H100**, sglang completes all queries in **≈18 seconds**, which is **~20%** of the total training iteration time.
>
> ### **Cost-normalized Accuracy**
>
> Figure 2 in the paper already presents a TFLOPs-normalized view of the trade-off (efficiency vs. accuracy).
> We additionally provide a structured comparison below, showing:
>
> * accuracy
> * per-card throughput
> * TFLOPs
> * memory requirement (in terms of GPU count)
>
> ---
>
> #### **Table 1. Efficiency–Performance Trade-off under Self-Ensemble**
>
> #### **EditScore-7B (host: 1×H100)**
>
> | K | Accuracy | Throughput (samples/s per GPU) | TFLOPs | Min GPU Requirement |
> | - | -------- | ------------------------------ | ------ | ------------------- |
> | 1 | 0.659    | 9.46                           | 36.68  | 1×H100              |
> | 2 | 0.714    | 6.58                           | 73.36  | 1×H100              |
> | 4 | 0.727    | 4.06                           | 146.72 | 1×H100              |
>
> ---
>
> #### **EditScore-32B (host: 2×H100)**
>
> | K | Accuracy | Throughput (samples/s per GPU) | TFLOPs | Min GPU Requirement |
> | - | -------- | ------------------------------ | ------ | ------------------- |
> | 1 | 0.680    | 2.37                           | 153.4  | 2×H100              |
> | 2 | 0.715    | 1.58                           | 306.8  | 2×H100              |
> | 4 | 0.733    | 0.96                           | 613.6  | 2×H100              |
>
> ---
>
> #### **EditScore-72B (host: 4×H100)**
>
> | K | Accuracy | Throughput (samples/s per GPU) | TFLOPs | Min GPU Requirement |
> | - | -------- | ------------------------------ | ------ | ------------------- |
> | 1 | 0.703    | 0.97                           | 328.5  | 4×H100              |
> | 2 | 0.757    | 0.635                          | 657    | 4×H100              |
> | 4 | 0.763    | 0.40                           | 1314   | 4×H100              |
>
> ---
>
> #### **Result Analysis**
>
> 1. **Self-ensemble improves accuracy consistently across all model sizes.**
>    For example, EditScore-7B increases from **0.659 → 0.727** (+6.8%) when K grows to 4.
>
> 2. **Cost grows sublinearly when K increases**, thanks to:
>
>    * shared prefill stage
>    * parallelization within sglang
>    * no increase in memory or model hosting cost
>
> 3. **Small models benefit most under equal hardware budgets.**
>
>    * On a *single* H100, **7B-avg4 (Acc=0.727)** outperforms **32B-avg1 (Acc=0.680)**,
>      despite the latter requiring double the hardware.
>      → Under cost-normalized settings, **7B with self-ensemble achieves the best accuracy–efficiency balance**.
>
> 4. **Larger models show diminishing returns**:
>
>    * 72B-avg4 improves only marginally over 72B-avg2 (0.763 vs 0.757).
>
> 5. Overall, the analysis **confirms that self-ensemble does not disproportionately increase inference cost**, while significantly improving reward accuracy.

---

> ### Author Response · Authors · 2025-11-24
> **Official Response to Reviewer bWV1 (Part  4 of 4)**
>
> > **W5: Claims about reward variance aiding optimization are intriguing but not causally established; isolate variance from annotator accuracy with controlled studies and replicate the effect on additional datasets and editors.**
>
> **A5**: Thank you for the valuable feedback. We agree that fully establishing the *causal* role of reward variance requires more controlled experimentation. Our current findings demonstrate a strong empirical correlation: reward design choices that increase variance consistently improve RL stability and editing quality. However, isolating variance from annotator accuracy is non-trivial, and developing principled methods for manipulating variance remains an open research question.
>
> We view our work as an initial step toward this goal. Prior work [3] has suggested similar limitations of low reward variance in language models, but actionable strategies are still lacking. Our study extends this observation to the **visual editing domain**, providing initial evidence that reward variance can influence RL performance.
>
> In future work, we plan to (1) conduct controlled studies that explicitly fix annotator accuracy while varying reward variance, (2) replicate the analysis across additional datasets and editing models, and (3) explore causal interventions that can isolate and quantify the effect of reward variance more rigorously. We believe these directions will help establish a clearer causal understanding and contribute to building stronger visual reward models.
>
> [3] *What makes a reward model a good teacher? an optimization perspective*, NeurIPS 2025.
>
> > **Q1: Benchmark: How were prompts/edit types sampled and how representative are they?**
>
> **A6**: Thank you for the question. Our benchmark is constructed using a stratified sampling strategy to ensure broad coverage and representativeness. We organize all instructions into **four major categories comprising 13 subtasks**, curated from widely adopted benchmarks such as **GEdit-Bench-EN** and **ImgEdit-Bench**, which have been empirically shown to cover diverse real-world editing needs.
>
> * **Subject-level edits** (addition, removal, replacement): fundamental structural manipulations frequently seen in practical editing pipelines.
> * **Appearance-level edits** (color/material change, style transfer, tone adjustment): shape-preserving modifications requiring fine-grained visual control.
> * **Scene-level edits** (background change, extraction): tasks demanding global spatial understanding and coherence.
> * **Advanced edits** (text editing, portrait beautification, motion change, hybrid edits): complex cases requiring higher-level reasoning or multi-attribute transformation.
>
> We sample images and instructions proportionally across these subtasks, ensuring a balanced mix of both common edits and more challenging scenarios. This design spans a wide range of “edit distances,” from small attribute tweaks to large structural transformations, resulting in a benchmark that reflects the diversity of realistic image-editing instructions encountered in practice.
>
> > **Q2: Raters: What was raw disagreement and how do results change without consensus filtering?**
>
> **A7**: Thank you for the question. Before finalizing our annotation protocol, we conducted an initial phase where multiple annotators labeled a pilot set independently. During this stage, most raw disagreements were resolved through discussion and justification, which helped us establish a stable and reliable evaluation rubric.
>
> For the full benchmark, we adopted the **Two-Annotator Discussion Protocol**, where two annotators jointly inspect and discuss each pair to reach a final decision. **The full disagreement analysis for this protocol is provided in our response to W1**, where we compare single-annotator vs. two-annotator settings in terms of convergence rate and agreement with majority.
>
> Importantly, **we do not apply any consensus filtering**: no samples are removed due to disagreement, and the two-annotator decisions directly form the benchmark labels. As shown in the disagreement analysis, the protocol achieves consistently high agreement, indicating that the resulting annotations are reliable and reflective of true human preference.
>
> For clarity and transparency, we have added a detailed description of our annotation pipeline in **Section 3.3 (Annotation Pipeline)** of the revised paper, with all changes highlighted in blue.

---

> ### Author Response · Authors · 2025-11-27
> **Follow-up Comment and Request for Updated Assessment**
>
> Thank you again for your thoughtful and constructive comments. We would like to kindly bring your attention to our detailed responses in the rebuttal, where we have addressed each of your concerns point-by-point and provided substantial additional experiments, including:
>
> - Controlled disagreement analyses and clarifications on the annotation protocol;
> - Release plans for the benchmark, RM training data, and RL training data;
> - Ablation studies across multiple editors, RL methods (including TempFlow-GRPO), and OOD stress tests;
> - A comprehensive efficiency–performance analysis for the self-ensemble strategy;
> - Additional discussions and future directions regarding reward variance.
>
> **We sincerely hope that these updates resolve your concerns. If our clarifications and new experiments satisfactorily address the questions you raised, we would greatly appreciate your consideration for a higher score.**
>
> Thank you very much for your time and constructive feedback.

---

> ### Author Response · Authors · 2025-12-03
> **Follow-up: Hyperparameter Ablation Results**
>
> Dear Reviewer bWV1,
>
> Following up on our previous response regarding the **"Influence of Hyperparameters,"** we have completed the promised
> ablation studies during the rebuttal period. We analyze the impact of (1) the number of inference steps and (2) the number of samples per prompt (rollouts) during RL training.
>
> The results and our analysis are presented below.
>
> ### 1. Impact of Inference Steps
>
> We evaluated the model performance with 12, 16, and 20 inference steps. As shown in the table below, increasing the number of inference steps leads to consistent improvements across all metrics (SC, PQ, and O) on GEdit.
>
> **Table 1: Ablation study on the number of inference steps on GEdit.**
> | Inference Steps | SC | PQ | O |
> | :---: | :---: | :---: | :---: |
> | 12 | 7.35 | 7.75 | 6.98 |
> | 16 | 7.42 | 7.80 | 7.11 |
> | **20** | **7.53** | **7.99** | **7.21** |
>
> *Analysis:* The performance gain with more inference steps is expected. Additional steps allow the flow model to generate higher-quality samples that better reflect its true distribution, enabling more reliable reward signals. With too few steps, artifacts appear and degrade reward evaluation.
>
> ### 2. Impact of Samples per Prompt (Rollouts)
>
> We further investigated the trade-off between the number of rollouts per prompt and the diversity of prompts within a batch. To ensure a fair comparison, we kept the **total computational budget (total batch size) constant**. Consequently, a higher number of rollouts corresponds to fewer unique samples (prompts) per update.
>
> **Table 2: Ablation study on the number of rollouts per prompt on GEdit.**
> | Unique Samples | Rollouts | SC | PQ | O |
> | :---: | :---: | :---: | :---: | :---: |
> | 36 | 8 | 7.36 | 7.89 | 7.07 |
> | **24** | **12** | **7.53** | **7.99** | **7.21** |
> | 18 | 16 | 7.26 | 7.80 | 6.90 |
>
> *Analysis:*
> We observe a "sweet spot" at **12 rollouts**:
> *   **Too few rollouts (8):** While the model sees more unique prompts (72), the advantage estimation for each prompt may be less accurate, leading to suboptimal updates.
> *   **Too many rollouts (16):** Although the estimation per prompt is stable, the number of unique samples (36) is significantly reduced. This lack of batch diversity appears to hinder the model's generalization ability, causing a performance drop (O: 6.90).
> *   **Optimal Setting (12):** This configuration provides the best balance between advantage estimation stability and data diversity.
>
> These additional experiments on GEdit further validate our parameter choices and demonstrate the robustness of our reward model. We will include these detailed ablation studies in the final version of the paper.
>
> Thank you again for your constructive feedback.

---

### Author Response · Authors · 2025-11-24
**Global Response to All Reviewers and AC**

We thank all reviewers for their time and constructive feedback. We are encouraged by the recognition of our work’s **originality (R1)**, **experimental quality (R1, R4)**, and **practical significance (R2, R3)**.

During the rebuttal period, we have conducted extensive additional experiments and analyses to address the reviewers' concerns, particularly regarding **generalization, benchmark rigor, and efficiency**. Below is a summary of our major revisions and response efforts.

### **1. Summary of New Experiments & Revisions**

To address concerns about the scope and robustness of our method, we have added the following experiments:

* **Generalization to Other Editors (R1, R2, R3):**
    We extended our RL experiments to **Flux-Kontext-dev**. Results show that EditScore consistently improves performance across all metrics (e.g., **+0.57 SC** and **+0.34 PQ** on GEdit-Bench), proving that our reward model (RM) is effective across different model architectures, not just OmniGen2.
* **Generalization to Other RL Algorithms (R1):**
    We validated EditScore with **TempFlow-GRPO** (arXiv:2508.04324). The results confirm that our RM provides high-quality signals that further boost performance when combined with advanced RL optimization techniques.
* **Generalization to Out-of-Distribution Data (R1, R4):**
    We performed stress tests on the **EmuEdit** benchmark (OOD tasks). EditScore-guided RL achieved consistent gains (e.g., **CLIP-I 0.876 $\to$ 0.900**), demonstrating strong generalization beyond the training distribution.
* **Controlled Baselines for Reward Modeling (R4):**
    To isolate the contribution of the RM, we compared EditScore against GPT-4.1 and Qwen2.5-VL baselines using the *same* base editing model. EditScore consistently outperformed baselines, confirming that the gains stem from a better reward signal.
* **Statistical Significance Testing (R3):**
    We conducted **paired bootstrap tests (10,000 rounds)**. All improvements over baselines were found to be statistically significant with $p < 0.001$.
* **Efficiency Analysis (R1, R3):**
    We provided a detailed cost-performance analysis. We clarified that with shared prefill in serving frameworks (e.g., sglang), the **self-ensemble strategy (Avg@4)** incurs **sublinear latency growth** (not 4x), making it practical for large-scale training.
* **Benchmark Rigor & Data Release (R1, R2, R3):**
    * We conducted a **controlled disagreement study**, showing that our **Two-Annotator Discussion Protocol** achieves near-perfect agreement (>95%) compared to single annotators.
    * We clarified our rigorous **de-duplication pipeline** (pHash, CLIP, Semantic filtering) to prevent data leakage.
    * **Crucially, we committed to open-sourcing the Benchmark, RM training data, RL training data, Reward Mode, Training Code, fine-tuned Editing Models** to facilitate reproducibility.

### **2. Restating Core Contributions**

We believe our work addresses a critical gap in the field of instructional image editing:

1.  **First Comprehensive Reward Benchmark:** We propose **EditReward-Bench**, a high-quality, human-annotated benchmark specifically designed to evaluate reward models for editing.
2.  **Robust Reward Model (EditScore):** We develop a family of generative reward models built on meticulous data curation and a self-ensemble strategy, achieving much stronger human alignment than existing open-source VLMs and rivaling proprietary ones.
3.  **Unlocking Online RL for Editing:** We demonstrate that EditScore enables stable and scalable online RL, significantly improving the performance of state-of-the-art editors.
4. **Fully Open Source**: We will release **all data, models, code, and the benchmark**, providing the first complete open-source pipeline from evaluation to reward modeling to online RL for image editing.

We are committed to incorporating all the valuable feedback and the new experimental results into the final version of the manuscript. We hope these efforts resolve the remaining concerns, and we look forward to further discussion.

---

### Author Response · Authors · 2025-12-03
**Summary of Rebuttal & Discussion to AC**

**To the Area Chair:**

We understand the unique challenges of this review process. We are writing to provide a concise summary of our rebuttal updates. During the rebuttal period, we conducted extensive additional experiments and provided detailed, point-by-point answers to every concern.

Importantly, **Reviewer tH54 engaged actively, acknowledged our responses addressed most concerns, and raised their score from 4 → 6 on Nov 26, 21:34 EST (before double-blind leak event)**. This demonstrates that the rebuttal substantially improved reviewers’ confidence. Although scores have now been reverted due to the double-blind leak event, we hope the AC can consider the intended score updates and the documented discussion.

Below, we summarize how our rebuttal resolved each reviewer’s concern, and why we believe the paper meets the ICLR acceptance bar. If you are interested in detail, *please refer to our responses to each reviewer*. The revised paper, with changes highlighted in blue, has been uploaded to OpenReview.

### **1. How Our Rebuttal Directly Addresses Reviewers’ Concerns**

- **Benchmark Rigor & Data Quality (R1, R2, R3) — Addressed (Sec. 3.3 and App. B)**

	We added:
	* A **controlled disagreement study**, showing that our **Two-Annotator Discussion Protocol** achieves near-perfect agreement (>95%) compared to single annotators.
	* We clarified our rigorous **de-duplication pipeline** (pHash, CLIP, Semantic filtering) to prevent data leakage.

- **Generalization to Other Editors (R1, R2, R3) — Addressed (Tab. 5.(a))**

	We extended RL experiments to **Flux-Kontext-dev**, showing consistent and significant gains (e.g., **+0.57 SC**, **+0.34 PQ** on GEdit-Bench).
	→ Demonstrates our reward model (EditScore) generalizes beyond OmniGen2.

- **Generalization to Other RL Algorithms (R1) — Addressed (Tab. 5.(b))**

	We validated EditScore using **TempFlow-GRPO**.
	→ Confirms EditScore provides high-quality and sTab. reward signals regardless of the underlying RL algorithms.

- **Ablation studies on Hyperparamters of RL algorithms (R1) — Addressed (in App. E)**

	We conducted ablations on key RL hyperparameters.
	→ Confirms the robustness of our RL framework and validates our chosen settings.

- **Out-of-Distribution Robustness (R1, R4) — Addressed (Tab. 4)**

	We tested on the **EmuEdit OOD benchmark**, showing clear performance improvements (e.g., **CLIP-I 0.876 → 0.900**).
	→ Demonstrates generalization beyond training distribution.

- **Efficiency / Practicality of self-ensemble strategy of our reward model (R1, R3) — Addressed (in Sec. 5.2 and Fig. 2.(b))**

	Efficiency analysis added: With sglang, **Avg@4** incurs sublinear overhead (EditScore-7B throughput: **37.84** for avg@1 vs. **16.24** for avg@4). This $\sim$2.3$\times$ factor—significantly less than the theoretical 4$\times$—confirms practicality for real-world deployment (details in App. C).
- **Controlled Reward Model Baselines (R4) — Addressed (Tab. 4)**

	We compared EditScore to GPT-4.1 and Qwen2.5-VL using the *same editor*.
	→ EditScore consistently outperforms both, isolating the contribution of the reward model.
- **Statistical Significance (R3) — Addressed (in App. D)**

	We ran **paired bootstrap tests (10,000 rounds)**.
	→ All improvements in Tab. 2 and Tab. 5 are statistically significant at **p < 0.001**.
- **Limitation Discussion (R3) — Addressed (App. L)**

	We expanded the limitation discussion in the revised paper and added concrete, illustrative cases in App. I, clearly outlining the current boundaries of our method and where future improvements are needed.
- **Reproducibility Guarantee (R1, R2) — Addressed**

	We clarified that we will release **the entire pipeline**:
	* Benchmark
	* Reward model
	* RM training data
	* RL training data
	* Code and fine-tuned editors

	→ Ensures full reproducibility and community benefit.

### **2. Why the Paper Meets the Acceptance Standard**
Our work fills a critical missing component in instruction-guided image editing: **a reliable, high-quality reward model and benchmark**, enabling the first practical, stable and reproducible open-source online RL pipeline for image editing.

**Key contributions reaffirmed:**
1. **A comprehensive benchmark for editing (EditReward-Bench).**
2. **High-quality generative reward model (EditScore) rivaling proprietary systems.**
3. **Consistent RL improvements across editors, RL algorithms, and benchmarks.**
4. **Full open-source release**, providing immediate utility to the research community.

Given that:
* Reviewer tH54 explicitly changed their score from 4 → 6 after discussions,
* Most critical concerns could be thoroughly addressed with new experiments and evidence,
* The work introduces infrastructure that the community currently lacks,

we believe the paper is **above the acceptance threshold**, even under the reverted-score situation.

We sincerely appreciate your effort in this unusual review cycle and thank you for your careful consideration.

---

### Meta-Review · Area_Chair_xsyV · 2026-01-07

**Summary:**

This paper received two negative initial reviews: score 4 from reviewer tH54, score 4 from rtMd.

The major concerns from reviewers include:
--use of candidate images from current models
--experiment setting in section 6.4.1
--some technical clarity like score in table 2, or the aggregation scores
--limited RL Scope
--computational Cost
--comparison with More Baselines
--Clarity and justification on annotation, some experiments like table 2 or 5
--failure mode and limitation

**Reviewer Concerns:**

Most concerns have been addressed.

**Reviewer Scores:**

tH54 mentioned they will raise the score to 6. And I believe rtMd might also increase the score.

---

### Decision · Program_Chairs · 2026-01-26

Accept (Poster)